# Calculating fast differential genome coverages among metagenomic sources using micov
Yuhan Weng [1,2,17], Caitlin Guccione [1,2,3,17], Daniel McDonald [2,17], Renee Oles [2,4], Suzanne Devkota [5], Evguenia Kopylova [6], Gregory D. Sepich-Poore [7,8], Rodolfo A. Salido [2], M. Omar Din [2], Se Jin Song [9], Kit Curtius [3,10,11], Hiutung Chu [12,13], Andrew Bartko [2,7,9], Jeff Hasty [7,14] & Rob Knight [2,7,9,15,16] ✉

Breadth of coverage, the proportion of a reference genome covered by at least one sequencing read, is critical for interpreting metagenomic data, informing analyses from genome assembly to taxonomic profiling. However, existing tools typically summarize coverage breadth at the whole-genome or aggregate-sample level, missing informative variation along genomes and between sample groups. Here we introduce MIcrobiome COVerage (micov), a tool that computes and compares per-sample breadth of coverage across many genomes and samples. micov offers two key advances: (1) rapid cumulative coverage breadth calculations specific to each sample type, and (2) detection of differential coverage breadth along genomes. Applying micov to three metagenomic datasets, we show that it identifies a genomic region in *Prevotella copri* that explains variation in community composition independent of host country of origin, uncovers dietary association with a partially annotated region in an uncharacterized *Lachnospiraceae* genome, enabling hypothesis generation for genes of unknown function, and improves sensitivity in low-biomass settings by detecting a single genomic copy of enteropathogenic *Escherichia coli* (EPEC) in wastewater and distinguishing *Mediterraneibacter gnavus* across specimen types.

Breadth of coverage—the proportion of a reference genome covered by at least one sequencing read—underpins the accuracy and precision of many bioinformatic analyses, from assembly to taxonomic profiling[1–3]. It is a quantitative, aggregatable statistic of sequencing data, and can therefore identify biologically-informative differential regions (e.g., bacterial strain variation) that differ between hosts with different characteristics. However, current tools focus on single measures of coverage breadth across all samples, such as mean coverage or aggregate coverage. They therefore do not exploit variation in coverage breadth along genomes and between samples[1,2].

To address this issue, we developed MIcrobiome COVerage (micov), a bioinformatic tool that computes precise, per-sample breadth of coverage across arbitrary numbers of genomes and samples, performs aggregate calculations, and indexes this information for fast look-up. micov tool can therefore rapidly assess differential coverage breadth between groups of samples by using cumulative coverage breadth calculations. These comparative analyses reveal associations between breadth of coverage patterns and sample metadata, including phenotypic traits and environmental variables, that cannot be seen when all samples are combined.

We illustrate the importance of metadata-specific aggregation using three representative metagenomic applications. First, we identified a genomic region in *P. copri* with a greater and independent effect on overall microbiome composition than the known large effect of country of origin. Second, we uncovered a novel dietary association with a partially annotated region in an uncharacterized *Lachnospiraceae* taxon, which enabled hypothesis generation for genes with unknown functions in an annotation-independent manner. Finally, we showed that we can detect a single genomic copy of EPEC in wastewater, and also differentiate the presence of *M. gnavus* in distinct types of low biomass specimens. These applications demonstrate how micov quantitatively detects associations between microbial community composition and specific genomic regions that cannot be seen when all the data are aggregated, identifies associations between specific genomic regions and phenotypic traits, and improves the detection of taxa of interest in low-biomass settings. Throughout this article, we use 'coverage' to refer specifically to breadth of coverage (fraction of reference genome covered by at least one read), not depth of coverage (average reads per base), unless otherwise specified.

## Results

### micov enables cumulative and position-based coverage visualization

micov tool processes Sequence Alignment/Map (SAM) files, allowing the user the flexibility to choose parameter settings such as match threshold and algorithm upstream of micov. It then produces per-sample, per-genome coverage intervals. Users can visualize coverage by sample for specified genome or genomic regions through cumulative and position coverage plots. Cumulative coverage plots rank samples within a metadata group

from least to greatest coverage, then plot the cumulative coverage of each sample together with all preceding ones. The x-axis represents samples ordered from lowest to highest genome coverage breadth, and the y-axis shows the cumulative coverage breadth. (Fig. 1). The idea for cumulative coverage came from multiple exposure photography in astronomy. In any given photograph, the number of photons of light from a faint object is impossible to distinguish from background noise. However, repeated observations of the same object accumulate enough signal in the same place to distinguish the object from background noise. Extending this idea to

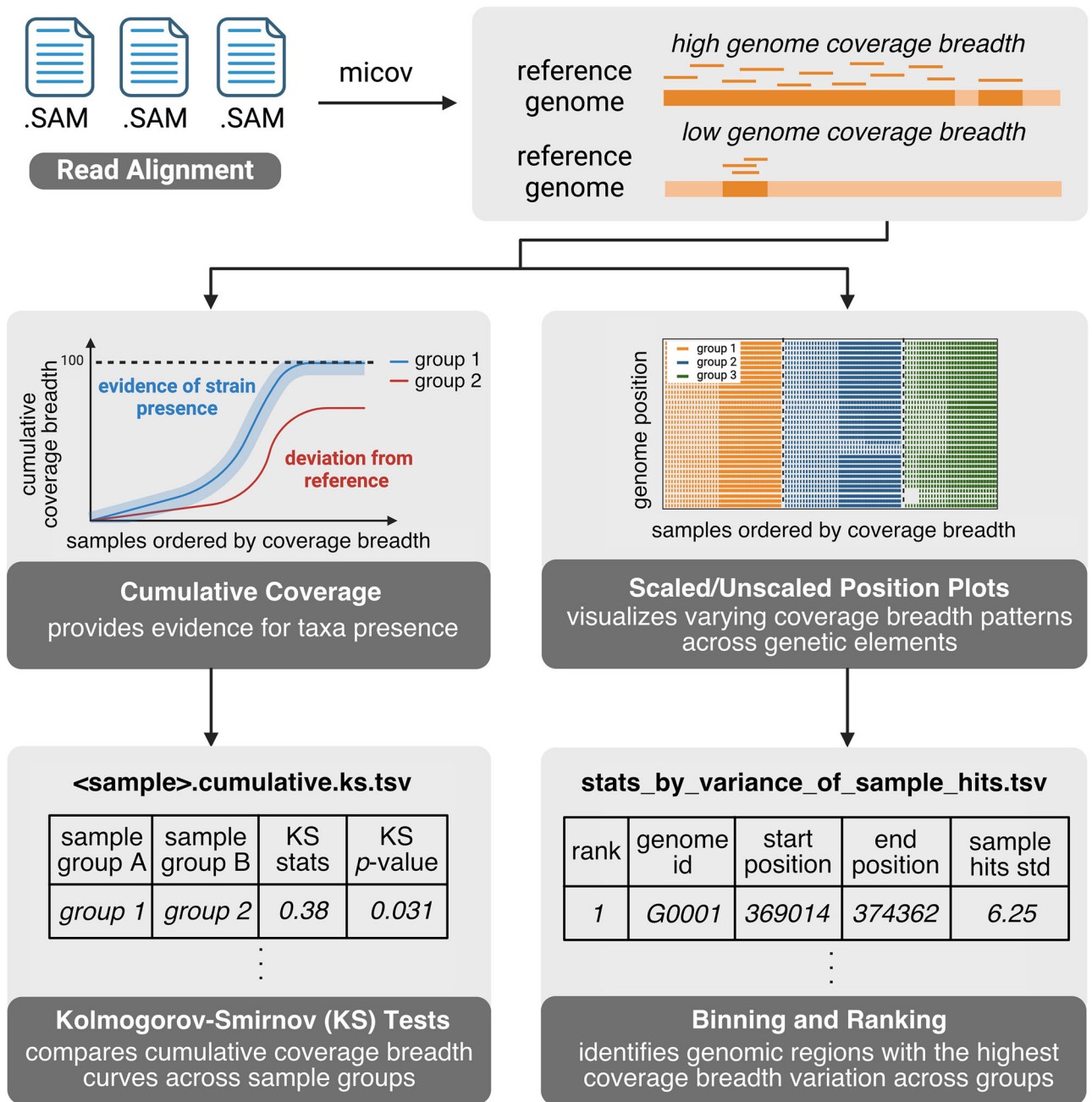

**Fig. 1 | Schematic of the micov workflow.** Cumulative coverage represents the total coverage breadth achieved when samples are ordered by increasing individual coverage breadth and added sequentially. This display helps assess whether coverage continues to accumulate across a sample group, or instead plateaus early to suggest that only a small part of the genome is represented in the sample. Position plots display coverage patterns across a reference genome. After cumulative coverage plots are generated, a Kolmogorov-Smirnov (KS) test is conducted to quantify differences between chosen sample groups, especially when visual inspection is challenging due to overlapping curves. Key columns of the output include sample groups being compared, KS statistic, and p-value. For genomic region variation, the genome is divided into *N* bins, and regions are ranked based on the standard deviation (std) of sample hits across groups. Key columns of the output include the ranking, genome id, start and end position of the genomic region, and standard deviation of sample hits (Methods). The figure is created in BioRender. Knight Lab (2024). https://BioRender.com/.

metagenomics, we reasoned that it might be impossible to distinguish a signal stemming from a small number of sequences matching a genome in one sample from background noise, but that if the genome were truly present then accumulation of the genome across many samples should yield a pattern of random accumulation over the full length of the genome. Furthermore, if the genome were present in some categories of samples but not others (for example, present in stool from individuals with a specific disease but absent from individuals without that disease), then the accumulation curves would differ in sets of samples grouped according to these categories.

Accordingly, our first use of cumulative coverage was an unpublished application for gathering evidence of the presence of a genome similar to a reference from a large collection of low-biomass metagenomic samples, stratified by clinical metadata. The intuition was that we can reasonably assume an individual sample will have a handful of reads approximately uniformly distributed over the genome if it is present, but the small counts may not be distinguishable from background. If coverage increases randomly across the genome when adding samples, especially if coverage only increases with samples within a specific category, this adds support to the hypothesis that this particular genome or a related one is present in the sample. This pattern would also rule out alternative hypotheses, such as the hypothesis that the matches are driven by lateral transfer to another organism or convergent evolution (accumulation would saturate at coverage of only a small part of the genome), contamination (accumulation would be unrelated to sample category), etc. Cumulative plots can include null models from Monte Carlo simulations, with cumulative curves tested for significance using pairwise Kolmogorov-Smirnov (KS) tests.

Position plots illuminate patterns of coverage across samples stratified by sample metadata. A scaled variant accommodates sparse data (Fig. 1). micov can bin genome regions to identify variable coverage across sample groups. Variables describing the presence/absence of coverage in these bins can be extracted for subsequent analysis (Methods). Collectively, these capabilities help assess genetic element distribution and relate coverage patterns to phenotypes.

## Micov highlights strain heterogeneity, distinguishing sample groups

Using this framework, we first applied micov to The Human Diet and Microbiome Initiative (THDMI) subset of The Microsetta Initiative[4], comprising 1218 usable human fecal metagenomic samples from the United States (US, $n = 412$), United Kingdom (UK, $n = 309$) and Mexico ($n = 497$). We focused on *Prevotella copri* (*Segatella copri*) based on its high prevalence (all samples contained some reads matching *P. copri*, Fig. 2a,b) and known strain variation[5]. We examined the top 10 genomic regions of variation as detected by micov's binning (Methods). Of these, the region spanning coordinates 351,299-354,812 ("PC351") was notable. PERMANOVA of weighted UniFrac distances indicated that presence/absence of PC351 alone exhibits a stronger effect on the overall microbiome composition than country of origin in the THDMI metagenomic data (Fig. 2c). This analysis was also supported by paired 16S rRNA data from the same samples (Supplementary Fig. S1a). Importantly, because PC351 contains protein-coding rather than ribosomal genes, the 16S rRNA data cannot directly provide information about this region. Unexpectedly, accounting for region PC351 when calculating PERMANOVA increased the effect size of variables such as antibiotic history (Fig. 2c; 16S rRNA in Supplementary Fig. S1a). Replicating this observation, PC351 separates samples in principal coordinates analysis better than country (Fig. 2d; 16S rRNA data in Supplementary Fig. S1b), with visually distinct separation (Fig. 2e; 16S in Supplementary Fig. S1c). Comparing cumulative coverage across groups, we observed a continued increase in genome breadth even in samples lacking the PC351 region (Fig. 2b). Additionally, the coverage patterns along the genome were qualitatively similar between groups with and without PC351 (Fig. 2a), hinting at the possibility that *P. copri* detection in samples lacking PC351 may not be a false positive (Fig. 2b). This is consistent with previous findings, because *P. copri* is a well-known human gut microbiome

component[6]. Although individuals with PC351 tended to have higher coverage of the genome than those who lacked the region (Fig. 2a), that a single potentially polymorphic genomic region would be associated with such a large difference in the microbiome overall was intriguing.

We then hypothesized that microbiome composition could predict the presence of this region, which we evaluated using a nested crossfold Random Forest classifier. The resulting classifier was highly accurate, with an area under the ROC curve (AUROC) of 0.91 (Fig. 2f; 16S rRNA data in Supplementary Fig. S1d). Although a classifier constructed to predict an individual's country had a larger AUROC of 0.97 (Supplementary Fig. S1e), we found *P. copri* to be ranked 679th in feature importances for country prediction, showing that the result is specific to this region of the *P. copri* genome, not to the species overall. Similarly, the resulting feature importances for country and PC351 were not correlated (Spearman, $\rho = 0.036$, $p = 0.08$), indicating that these classifiers detect different underlying signals. Finally, we searched the sequence for PC351 by BLAST[7], observing 100% identity to two independent records (CP085932.1, CP102288.1) containing a gate domain containing protein (UniRef A0A229I1P7). That annotation suggests that the protein has an extracellular role, and prompts follow-up studies of its effects on microbial interactions.

## micov uncovers associations linking genetic elements to phenotypic traits

Next, we examined genomes linked to the diversity of plant consumption, which had a large effect on the microbiome in subjects participating in Microsetta[4,8,9]. The genome and region with the greatest variation associated with different types of plants consumed per week was an unnamed *Lachnospiraceae*, at coordinates 682,000-695,000 ("L682"). This region exhibited an associated coverage differential with higher coverage for subjects on a high-plant diet (>30 different plants) compared to those on a low-plant diet (<10 different plants) (Fig. 2g; Wilcoxon Rank-Sum Test, $U = 14,5245$, $p = 6.99^{-9}$). This observation suggests two hypotheses: (1) strains of unnamed *Lachnospiraceae* may be adapted to diets diverse in different fibers, and (2) diets that include consumption of diverse plant species may promote strain competition within this family. Notably, seven out of 15 predicted genes in this region associate with unknown functions across multiple annotation systems (Supplementary Fig. S1f; Supplementary Table 1). Therefore, micov's annotation-independent operation enables the generation of hypotheses for genes with unknown functions based on their associations with sample categories.

## micov aids taxonomic detection in low-biomass settings

We then applied micov to low microbial biomass environments. First, using wastewater from facilities in San Diego County and across the UC San Diego campus, we spiked in known genome copy counts of enteropathogenic *Escherichia coli* (EPEC). Using micov's cumulative coverage, we can detect EPEC spike-in of a single introduced genome copy relative to background wastewater data (Fig. 2h). Second, we examined microbial DNA from prospectively acquired, surgically resected, paired human mucosal and adipose tissue samples from patients with Crohn's disease, from a previous study. These tissues have already been extensively studied using a culturomics approach, identifying specific gut bacteria that consistently translocate to adipose tissue[10]. Using micov, we found that *Mediterraneibacter gnavus* (*Ruminococcus gnavus*)—a microbe associated with Crohn's disease[11] and capable of producing capsular polysaccharides that may aid tissue colonization[12]—exhibited significantly higher coverage in involved mucosa relative to other tissue types (Fig. 2i, KS test statistics in Fig. 2j). When we consider the coverage patterns along the genome, we further observe the potential for strain variation within and between tissue types (Supplementary Fig. S1g). Collectively, these examples demonstrate the sensitivity and utility of micov in low biomass settings.

Although existing coverage calculation tools like CoverM[13] and Anvi'o[14] support coverage computation and visualization, micov outputs

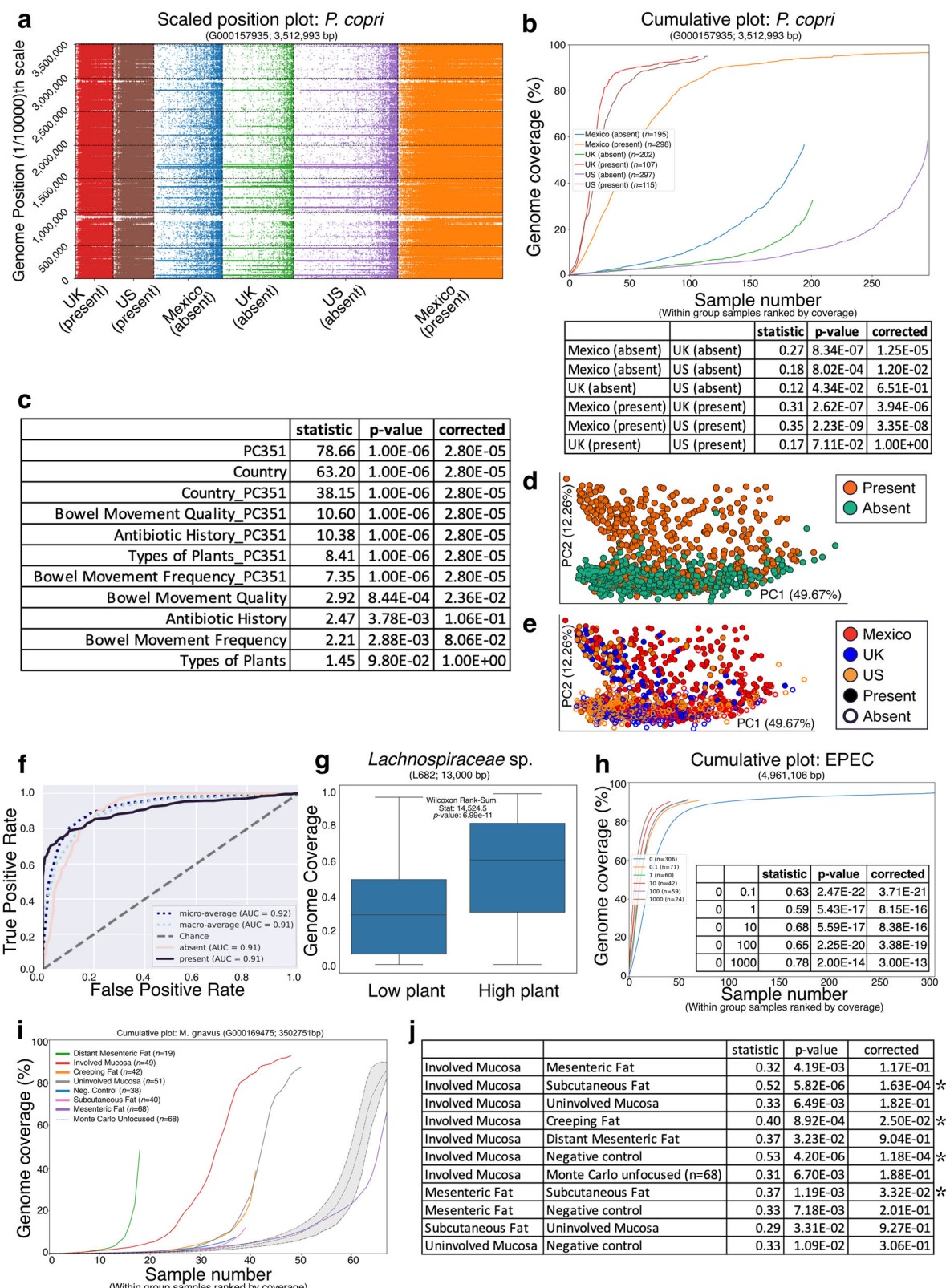

a table ranking genomic regions by decreasing coverage variation across sample groups. This functionality can highlight mobile elements, or other strain variations, that are specific to a subgroup of samples. micov also supports the computation of coverages at least as fast as the current state-of-the-art tool CoverM (Supplementary Fig. S2, Supplementary Table 3). micov's cumulative coverage calculations represent a new feature that, to

the best of our knowledge, is not available in other tools. This approach bypasses the selection of a coverage breadth threshold in detecting presence/absence of a particular microbial taxon in metagenomic samples (Supplementary Note 1). Although widely used tools such as InStrain[3], MIDAS2[15], and StrainPhlAn 4[16,17] excel at detecting single-nucleotide variations for high-resolution strain profiling, micov employs a coverage-

**Fig. 2 | micov detects phenotypically relevant strain variation, captures changes in genome abundance at the level of a single genomic copy in wastewater, and exhibits sensitive detection in low-biomass specimens. a** A scaled position plot of *P. copri* in human gut microbiome samples collected from subjects in the US/UK/Mexico stratified by presence/absence of region PC351. Sample groups are ordered by increasing sample size. Grey dotted gridlines are added as a visual esthetic to help understand the data relationship to the genome coordinates on the y-axis; **b** Coverage presence in this region is associated with greater overall genome coverage, with supporting Kolmogorov-Smirnov statistics. Notably, overall coverage was not significantly different between the US and UK for individuals containing the region (KS test, stat=0.17, $p = 0.0711$), nor was it if they both lacked the region (KS test, stat=0.12, $p = 0.0434$; n.s. if corrected); **c** Common high effect size variables, and per-sample characterization of region presence/absence, were tested with PER-MANOVA against Weighted UniFrac;PCoAs of the weighted UniFrac distances

colored by the region (**d**) and colored by country (**e**); **f** A receiver operator curve for a nested cross validated Random Forest classifier predicting presence/absence of PC351; **g** Coverage for region L682 in the *Lachnospiraceae* genome exhibiting differential coverage related to the diversity of plant consumption; **h** Detection of enteropathogenic *E. coli* (EPEC) at increasing levels of genome copies spiked into untreated wastewater (Methods). All spike-in levels show statistically significant elevated cumulative coverage levels compared to the background. A low background amount of *E. coli* is expected in wastewater. Only samples with non-zero EPEC coverage coverage ($n = 562$) are shown; **i** The cumulative coverage of *M. gnavus* from different tissue types surgically collected from Crohn's Disease patients; **j** with supporting Kolmogorov-Smirnov statistics. The set of statistics shown are those which reported an uncorrected $p$-value below 0.05, with correction by the Bonferroni procedure; asterisks denote corrected $p$-values below 0.05. All Kolmogorov-Smirnov and PERMANOVA tests are in Supplementary Table 2.

based approach that complements existing SNP-focused strain characterization tools to identify genomic segments with differential coverage patterns between sample groups.

Using micov, we identified a *P. copri* region with a greater and independent effect on microbiome composition than geographic country; this effect was recapitulated in 16S rRNA amplicon data from the same samples, which cannot observe the genomic region directly. We then uncovered a novel dietary association with a partially annotated region in an uncharacterized *Lachnospiraceae* taxon. Finally, we demonstrated sensitivity by detecting a single genomic copy of EPEC in wastewater, and separately, differentiating the presence of *M. gnavus* in distinct types of low-biomass specimens. In the latter, higher *M. gnavus* coverage in the mucosa than in fat likely reflects its passing through several selective sieves, meaning that only a small portion makes it to the adipose. In summary, micov unlocks a new layer of "coverage-omic" analyses for metagenomic datasets.

## Methods

micov is a BSD-3 licensed command-line Python program for computing coverage over genomes across samples, and comparing coverage among groups of samples. It consumes headerless SAM or BED3 data, and indexes the observed regions per sample per genome in Parquet[18] through a combination of Polars[19] and DuckDB[20]. Parquet is used specifically for its native support of out-of-core execution. micov can use genome lengths for coverage percent, accepts categorical sample metadata for plotting and differential testing, allows a user to express specific samples and genomes (or regions of genomes) to include in figure generation, supports automated region of interest discovery, and the extraction of regions of interest per-sample for use in downstream analysis. The tool produces key outputs including per-sample genome coverage tables, aggregated genome coverage across samples, cumulative genome coverage plots with supporting statistics, and position plots (scaled and unscaled). Users can use the outputs to guide the filtering of underrepresented genomes or exploring differential coverage across genomic regions.

### Genome Coverage Calculation

Each SAM file is parsed as a DataFrame using Polars version 1.21.0[19] with a schema corresponding to the general structure of a SAM file. The CIGAR string column is converted to the length of mapped regions using the following rule: if the CIGAR operation is among alignment match (M), sequence match (=), sequence mismatch (X), deletion from the reference (D), or skipped region from the reference (N), the corresponding numeric component of these operations is added to the length of covered region; if the CIGAR operation is among insertion to the reference (I), soft clipping (S), hard clipping (H), or padding (P), the total length of the covered region remains unchanged. The parsed SAM data are then reduced to a BED3-like DataFrame, which contains: the name of the reference genome, the start position, and the end position of the aligned read. The processed position information is compressed on the fly such that overlapping intervals are replaced with a single contiguous interval. This process (Supplementary

Fig. S3) converts non-numeric information stored in SAM files and CIGAR strings into a numeric table that tracks the start and stop positions with respect to the reference genome.

micov computes coverage using Woltka's CIGAR processing[21], inclusive of primary and secondary alignments so that true multimapping regions, such as copy number variants, horizontally transferred elements, and repeat regions, are represented in coverage calculations.

With the start and end positions, each contiguous interval is represented on the reference genome. The total length of these intervals, divided by the length of the genome, provides the estimated genome coverage of that reference genome.

### Cumulative coverage plots and null Monte Carlo distributions

Cumulative coverage plots are calculated by ranking each sample within its respective group by the percentage of observed genome coverage (or genome region if constraining). Specifically, the x-axis describes the per-group sample rank based on genome coverage. For each sample group, a curve is plotted, showing the total amount of coverage for that sample and all other samples with equal or lower coverage breadth, or, in other words, it represents the cumulative coverage.

For an optional null reference, a curve derived from a Monte Carlo procedure can be constructed. Specifically, the Monte Carlo procedure randomly permutes sample labels a user-determined number of times, for samples with greater than zero coverage for the genome ("focused" Monte Carlo), or for all samples ("unfocused" Monte Carlo), followed by computing the cumulative coverage. The number of samples selected is based on the largest sample group size.

Differences in cumulative curves are computed through pairwise Kolmogorov-Smirnov (KS) tests using SciPy[22] for all pairs of curves, including null Monte Carlo curves if present. For the cumulative plots in Fig. 2, we computed Bonferroni-corrected $p$-values based on the number of pairwise tests run for a specific analysis, but we only display the subset of results specific to the discussion. The full results are included in Supplementary Table 2.

### Position plots

Position plots visualize coverage variations across an entire genome or a region of a genome, depending on user parameters. The x-axis is split by sample groups. The groups are ranked from left to right by the number of samples in each group. Within a group, samples are sorted by the coverage of the genome or region. The y-axis represents the observed positions covered by sequencing data. The data are colored by the corresponding sample group. For a whole genome, the number of pixels within a vertical column is far less than the number of positions in the feature being examined; as a result, the visual may appear sparse for samples with low coverage.

In the case of only two sample groups (e.g., disease vs. control), the right-most group is ranked from greatest to least coverage so the samples with the highest coverage between both groups are horizontally adjacent.

## Scaled position plots

To improve position plot visualization, micov also produces a 1/10,000th scaled position plot. Specifically, a genome or region is represented by 10,000 bins, and a point on the y-axis corresponds to that bin containing at least some coverage for the given sample.

## Binning

Genomic regions with varying coverage patterns across sample groups are detected in micov by dividing the range of given genomes length into $N$ non-overlapping bins. A sample is considered present in the bin if at least one read from the sample is within the bins corresponding start and stop coordinates. The standard deviation of these counts, across sample groups, is computed. The resulting output is ranked by the standard deviation. Regions with high standard deviations represent areas with varying coverage patterns across sample groups.

## Installation

The micov package is BSD-licensed and is pip and conda installable. The code and installation instructions can be found at https://github.com/biocore/micov.

## THDMI data

The THDMI data previously published were used for analysis in this study[23]. Fecal samples were collected using the Microsetta Initiative platform[4] from 442 adult individuals in the U.S., 342 in the U.K., and 507 in Mexico, in accordance with the University of California San Diego (Protocol #: 141853) and the Instituto Nacional de Ciencias Medicas y Nutrición Salvador Zubiran (INCMNSZ) in Mexico (Protocol #: 3853). DNA was extracted using the MagMAX extraction kit as previously described[24]. Metagenomic libraries were prepared using the KAPA HyperPlus kit with miniaturized volumes as previously described[25], normalized using pre-sequencing on an Illumina iSeq platform[26], and sequenced on an Illumina NovaSeq 6000 at the Institute for Genomic Medicine at UC San Diego. Sequence files underwent quality control and human sequence filtering following the recommendations from Sepich-Poore et al.[27]. Briefly, adapters were trimmed using fastp version 0.23.4[28] and reads mapping to reference human genomes (HPRC[29], T2T-CHM13v2.0[30], and GRCh38[31]) were filtered using minimap2 version 2.26-r1175[32].

Amplicons of the V4 region of the 16S rRNA gene were generated from the same DNA extraction used for shotgun metagenomics using barcoded EMP primers[33]. Paired-end 150 bp reads were generated by sequencing the amplicons on an Illumina MiSeq at the Knight Lab[34].

The host filtered metagenomic sequence data were mapped against Web of Life version 2[35] using bowtie2 v2.3.2[36] within the standard processing pipeline in Qiita[37]. The resulting per-sample SAM files were compressed into BED3-like files using "micov compress" and converted into Parquet with "micov nonqiita_to_parquet". micov binning was performed for all genomes with "micov binning --bin-num 500". *Prevotella copri* (GCA_000157935.1; G000157935 in Web of Life[35] version 2; n.b., this taxon has been revised to *Segatella copri*[38], however, we opted for the use of its basionym for consistency with the reference database) was selected as a genome of interest given its high prevalence and significance in human microbiome literature[5]. We evaluated the top 12 ranked regions for *P. copri*, differentiating country, as identified through micov's binning, and focused on the region with the largest observed effect size as measured by PERMANOVA[39] over weighted UniFrac distances (n.b., PC351 was ranked 10th). The effect sizes of other regions from the top 12 can be found in Supplementary Table 2 (sheet: thdmi-wgs-weighted-raw). We then constructed per-sample variables describing whether the genomic region was present, absent (i.e, the sample had coverage for *P. copri* but not in this genomic region), or not applicable (i.e., the sample had zero coverage for *P. copri*). Four samples lacked detectable coverage of G000157935: 10317.X00214187, 10317.X00214884, 10317.X00215454, 10317.X00215765.

*Lachnospiraceae* (GCA_900066105.1, G900066105 in Web of Life version 2) was selected as a genome of interest based on its previously reported association with dietary plant diversity in multiple studies. Specifically, *Lachnospiraceae* exhibited differential abundance between individuals consuming fewer than 10 and more than 30 different plant types per week in the American Gut Project (AGP) cohort[4]. Additional studies have identified *Lachnospiraceae* as one of the top taxa associated with fermented plant consumption[9] and significantly enriched in plant-based diets compared to other diet types[8]. These consistent findings support its relevance in plant-associated gut microbiome responses and motivate our investigation into whether specific genomic regions within *Lachnospiraceae* correlate with dietary plant diversity.

16S rRNA V4 data were processed in Qiita[37] under study ID 10317 using the standard Deblur[40] v2021.09 processing trimmed at 100 nucleotides. A feature table of 16S rRNA data for the THDMI WGS sample IDs was obtained and processed through the Microsetta-processing project. Briefly, microsetta-processing obtained data using redbiom[41] v0.3.9 from context "Deblur_2021.09-Illumina-16S-V4-100nt-50b3a2" on December 14, 2024. Any ambiguities (e.g., a sample sequenced multiple times) were merged. Blooms were removed[42]. Samples with fewer than 800 sequences per sample were removed. ASVs were inserted into Greengenes 13.8[43] with SEPP[44] using q2-fragment-insertion with QIIME 2[45] 2022.2.

For analysis, using BIOM v2.1.15[46], 16S rRNA samples with fewer than 1000 sequences per sample, and metagenomic samples with fewer than 1,000,000 mapped reads, were removed from consideration. The set of sample IDs present in both preparation types was then obtained, and both the 16S rRNA and metagenomic feature tables were filtered to the set of common sample IDs. 1% coverage filter was applied prior to the 1 M filter and 16S overlap. The following four samples were excluded due to insufficient 16S data for the beta diversity analysis: 10317.X00215648, 10317.X00213982, 10317.X00215889, 10317.X00214932. These samples account for the difference between the current dataset and the larger superset of samples used in the previous THDMI publication[23]'s diversity analyses. Weighted normalized UniFrac[47] calculations with the unifrac-binaries v1.4.0 library for both the metagenomic and 16S rRNA V4 data were performed with the same phylogeny (Greengenes2 2024.09[48]). Rarefaction, principal coordinates and PERMANOVA were computed at the same time. 16S rRNA data were rarefied to 1,000 sequences per sample, whereas the metagenomic data were rarefied to 1,000,000 sequences per sample. PERMANOVA was computed using 999,999 permutations. The exact command was the following: "ssu -i <table>-subsample-depth <depth>-t <tree>-o -m weighted_normalized_fp32 -r hdf5_fp32 -pcoa 10 -g <metadata>-c <comma,separated,column,names,for,permanova>-permanova 999999". Bonferroni corrected *p*-values were computed for the PERMANOVA results in Fig. 2 and Supplementary Fig. S1 using the full set of variables tested at the time. These included other top-ranked regions identified by binning (Methods); their associated PERMANOVA statistics can be found in Supplementary Table 2 (sheet: thdmi-wgs-weighted-raw), and a presence/absence variable for detection of *P. copri*. The full table is included in Supplementary Table 2.

Random forest classifiers were constructed using q2-sample-classifier version amplicon-2024.5[37], with the "qiime sample-classifier classify-samples-ncv" command using default parameters.

## EPEC data

Enteropathogenic *E. coli* (EPEC) stocks were extracted using MagMax Microbiome Ultra Nucleic Acid Isolation kit adapted for magnetic bead purification as previously described[24]. The extraction process is automated to allow for the processing of many samples in parallel. Automation is achieved by the use of liquid handling and bead beating instruments. Nucleic acids were eluted in 100 μl MagMax elution buffer[24].

Frozen wastewater samples were thawed and transferred into 24-well plates containing Ceres Nanosciences Nanotrap Microbiome A Particles for microbial enrichment, followed by total nucleic acid extraction using MagMAX Microbiome Ultra Nucleic Acid Isolation kit with the automated KingFisher Flex liquid handler (ThermoFisher Scientific), as previously described[49]. Total nucleic acid was eluted in 65 μl MagMAX elution buffer.

Genomic copies present within the stock of extracted EPEC genomic DNA (gDNA) were determined via qPCR, where quantification was determined using a calibration curve generated by known amounts of genomic target loci. This gDNA stock was then used to generate wastewater spike-in samples with the following quantities: 0, 0.1, 1, 10, and 1000, genome copies. Spike-in's were performed at the level of extracted wastewater gDNA, and some control samples spiked-in to only water (154 samples), directly prior to sequencing library preparation. Water controls were excluded from Fig. 2h.

Shotgun metagenomic sequencing was performed using a 1:10 miniaturized KAPA HyperPlus (Roche) protocol as previously described[25]. Briefly, extracted DNA was quantified with PicoGreen dsDNA Assay Kit (ThermoFisher Scientific) and normalized to 5 ng of input or a maximum 3.5 µl of gDNA. Reagents for enzymatic fragmentation, end-repair and A-tailing, and adaptor ligation were added stepwise at miniaturized volumes using a Mosquito HV micropipetting robot (SPT Labtech). Magnetic bead washes associated with library prep were performed with the BlueWasher (BlueCatBio) and the Mosquito HV. Sequencing adapters and barcode indices were added in two steps, following the iTru barcoding strategy[50], and libraries were amplified for 15 cycles[25]. Individual libraries were pooled at equal volumes and sequenced on an Illumina iSeq 100 paired-end 150-cycle run. The total read count per sample from the iSeq results was used to normalize the pooling volumes for sequencing on an Illumina NovaSeq X Plus[26]. The pooled library was cleaned with the QIAquick PCR purification kit and size selected for fragments between 300 and 700 bp on the Sage Science PippinHT. The library was sequenced as a paired-end 150-cycle run on an Illumina NovaSeq X Plus at the UCSD IGM Genomics Center. We thus generated a set of 735 host-filtered sequencing samples derived from wastewater and control samples spiked with these varying levels of EPEC genome copies.

The EPEC reference genome was generated by PacBio sequencing on its extracted gDNA. We processed the EPEC using the HiFi plex prep kit 96, with a modified protocol to accommodate automated steps and shaking-based sample shearing, and sequenced using the Revio platform via the SMRTcell flow-cell. HiFi reads from PacBio sequencing were processed via Flye (v2.9.2)[51] based assembly using the Hybracter pipeline (v0.7.0)[52]. The assembled genome is available in Zenodo under DOI 10.5281/zenodo.15061184.

The 717 samples were aligned to the EPEC reference genome using bowtie2 v2.5.4[36] and samtools v1.21[53]. "micov compress" was used to compress the per-sample SAM output files to BED3-like files. Finally, all samples were combined into Parquet with "micov nonqiita-to-parquet". "micov per-sample-group" was run with the Parquet output against the EPEC reference genome using the metadata column, spike, denoting the number of EPEC genomes spiked into the sample: 0, 0.1, 1, 10, 100, or 1000 genomes.

## Visceral Fat Data

Shotgun metagenomics sequencing was performed on human mucosal and adipose tissue samples from Crohn's disease patients (tissue sampling methods and human subjects approval previously published[10]. The study was approved by the Cedars-Sinai Medical Center Institutional Review Board. Informed consent was obtained. All ethical regulations relevant to human research participants were followed, extracted using the MagMAX Microbiome Ultra Nucleic Acid Isolation kit in 96-well plates. Sequencing libraries were constructed with triplicate library replicates using the 1:10 miniaturized KAPA HyperPlus protocol described above. Individual libraries within each 384-well library prep, each plate being one technical replica, were pooled at equal volumes and sequenced on an Illumina iSeq 100 paired-end 150-cycle run. The total read count per sample from the iSeq results was used to normalize the pooling volumes for sequencing on NovaSeq X. Due to issues of elevated host contamination and low expected microbial biomass associated with the visceral fat sample set, sequencing libraries prepared in each technical replicate plate were split into sequencing pools of 18 samples and subsequently ultra-deep sequenced on a lane of a S4 flowcell of the NovaSeqX, totaling 47 lanes of ultra-deep sequencing effort. Sequencing pools were selected to batch samples with similar sequence performance evaluated by total read count from iSeq run and included no template controls (NTCs). After demultiplexing, samples were comprehensively host filtered[54]. Per-sample technical replicates were combined for analysis.

The host filtered metagenomic sequencing data were mapped against RS225 using bowtie2 v2.5.4[36] within the standard processing pipeline of Qiita[37]. The construction of the micov Parquet files was performed consistently with the THDMI processing noted above. *M. gnavus* (GCF_000169475.1) was picked as a focus for analysis given its prior relevance for the sample types being considered[11].

## Benchmarking

To evaluate performance, we benchmarked micov and CoverM across datasets of increasing size (from 100 to 10 million read alignments) randomly selected from the THDMI dataset to evaluate their runtime performance for coverage breadth calculation. All tests were performed using a single thread ($POLARS\_MAX\_THREADS = 1$ for micov, $-t$ 1 for CoverM) to avoid program-specific parallel overhead. We chose to focus our comparison on CoverM because its documentation explicitly states that it is a "fast DNA read coverage and relative abundance calculator focused on metagenomics applications". Although CoverM does not compute cumulative coverage breadth, its functionality can be constrained to compute per-sample, per-genome coverage breadth by using the following command: 'coverm genome --bam-files -m covered_fraction'. The covered fraction is a metric that micov also provides, making it a relevant and practical benchmark for evaluating runtime performance.

## Statistics and Reproducibility

PERMANOVA was computed using 999,999 permutations. The exact command was the following: "ssu -i <table>-subsample-depth <depth>-t <tree>-o -m weighted_normalized_fp32 -r hdf5_fp32 -pcoa 10 -g <metadata> -c <comma,separated,column,names,for,permanova>-permanova 999999". Bonferroni corrected $p$-values were computed for the PERMANOVA results in Fig. 2 and Supplementary Fig. S1 using the full set of variables tested at the time. Additional information on statistical analyses can be found in the relevant methods section.

## Reporting summary

Further information on research design is available in the Nature Portfolio Reporting Summary linked to this article.

## Data availability

Data from THDMI are part of Qiita study ID 10317 (Prep IDs: 16848, 16854, 16853, 16849, 16875, 16756, 16761, 16766) and European Bioinformatics Institute accession number PRJEB11419 (See Supplementary Data 1 for specific experiment accessions). Data from the EPEC study are available on EBI with accessions PRJEB80377 and ERP164391. The sequencing data is also available on Qiita with study ID 15666. The spike-in EPEC genome is available on EBI with accession GCA_976976645.1 and on Zenodo (https://doi.org/10.5281/zenodo.15061184)[55]. Data from the visceral fat study are available on EBI with accessions PRJEB98012 and ERP180450. Numerical source data for graphs and charts can be found in Supplementary Table 2. All other data are available from the corresponding author on reasonable request.

## Code availability

The software to perform coverage analysis is available under an open-source license and can be obtained at https://github.com/biocore/micov[56].

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

## Acknowledgements

This work was supported in part by AGA Research Foundation (AGA Research Scholar Award AGA2022-13-05) and NIH R01 CA270235 to K.C. The study was supported in part by the NIDDK-funded San Diego Digestive Diseases Research Center (P30 DK120515). Additionally, this work was supported in part by NIH grants (R01 CA241728, P30 CA023100, T32GM007198, DP1AT010885, U24CA248454, 5R24AI118629), and CDC award 75D301-22-C-14717 to R.K. R.K. and D.M. are supported by Danone Nutricia Research 191330. A.B. and S.S. are supported by Danone Nutricia Research 191330. The dataset from THDMI was generated through support from Danone Nutricia Research and the Center for Microbiome Innovation. The contents do not represent the views of the U.S. Department of Veterans Affairs or the United States Government.

## Author contributions

Y.W., C.G., and D.M. conceived and designed the study. Y.W., D.M., R.O., and R.A.S. processed the data. Y.W., C.G., D.M., R.O., and M.O.D. performed the primary analyses. Y.W., C.G., and D.M. wrote this manuscript with input from all authors. S.D., E.K., G.D.S., R.A.S., S.J.S., K.C., H.C., A.B., J.H., and R.K. aided in interpreting the results and drafting the manuscript. R.K. supervised the project. All authors reviewed and approved the final manuscript.

## Competing interests

The authors declare the following competing interests: D.M. is a consultant for BiomeSense, Inc., has equity, and receives income. E.K. is the managing director of Clarity Genomics. M.O.D. has equity in GenCirq. K.C. has research grant support from Phathom Pharmaceuticals. A.B. is a founder of Guilden Corporation and is an equity owner. R.K. is a scientific advisory board member and consultant for BiomeSense, Inc., has equity, and receives income. He is a scientific advisory board member and has equity in GenCirq. He is a consultant for DayTwo, and receives income. He has equity in and acts as a consultant for Cybele. He is a co-founder of Biota, Inc., and has equity. J.H. is a co-founder of GenCirq Inc., which focuses on cancer therapeutics. He is on the Board of Directors and has equity in GenCirq. His spouse is employed part-time for the bookkeeping and to support employees with Human Resources. The terms of these arrangements have been reviewed and approved by the University of California, San Diego, in accordance with its conflict of interest policies. The remaining authors declare no competing interests.

## Additional information

¹Bioinformatics and Systems Biology Program, University of California San Diego, La Jolla, CA, USA. ²Department of Pediatrics, University of California San Diego, La Jolla, CA, USA. ³Division of Biomedical Informatics, Department of Medicine, University of California San Diego, La Jolla, CA, USA. ⁴Department of Pathology, School of Medicine, University of California San Diego, San Diego, CA, USA. ⁵Human Microbiome Research Institute, Cedars-Sinai Medical Center, Los Angeles, CA, USA. ⁶Clarity Genomics, San Diego, CA, USA. ⁷Shu Chien-Gene Lay Department of Bioengineering, University of California San Diego, La Jolla, CA, USA. ⁸Feinberg School of Medicine, Northwestern University, Chicago, IL, USA. ⁹Center for Microbiome Innovation, University of California San Diego, La Jolla, CA, USA. ¹⁰VA San Diego Healthcare System, San Diego, CA, USA. ¹¹Moores Cancer Center, University of California San Diego, La Jolla, CA, USA. ¹²Department of Pathology, University of California, San Diego, La Jolla, CA, USA. ¹³Chiba University-UC San Diego Center for Mucosal Immunology, Allergy and Vaccines (cMAV), University of California, San Diego, La Jolla, CA, USA. ¹⁴Department of Molecular Biology, Division of Biological Science, University of California San Diego, La Jolla, CA, USA. ¹⁵Halıcıoğlu Data Science Institute, University of California San Diego, La Jolla, CA, USA. ¹⁶Department of Computer Science and Engineering, University of California San Diego, La Jolla, CA, USA. ¹⁷These authors contributed equally: Yuhan Weng, Caitlin Guccione, Daniel McDonald. ✉e-mail: rknight@ucsd.edu

