## [Transparent Peer Review file · Communications Biology]

Calculating fast differential genome coverages among metagenomic sources using micov

Corresponding Author: Dr Rob Knight

Version 0:

Reviewer comments:

Reviewer #1

(Remarks to the Author)

General comments:

In the paper "Calculating fast differential genome coverages among metagenomic sources using micov", Yuhan Weng and colleagues present a tool designed to compute and compare per-sample genome coverage positions across multiple genomes and samples.

While the concept behind the tool is not entirely novel, micov could prove useful when working with large-scale datasets involving numerous samples.

However, the paper is currently challenging to follow due to complex sentence structures and would benefit from a substantial reorganization, especially in the "Main" section. Simplifying the language would significantly improve readability. Additionally, there are several grammatical mistakes and formatting issues (e.g., in the references) that need to be addressed.

One of the more promising features of micov is its potential to investigate genomic variation across different samples for one or more species. Nevertheless, there is not a comparison with other tools performing strain characterization in metagenomic samples.

Also, the main target and intended use of the tool should be explained more clearly at the beginning—this only becomes apparent after reading a large portion of the manuscript. Clarifying these points and improving overall readability would significantly strengthen the paper.

Specific Comments:

Line 46: Consider referring to "breadth of coverage" instead of just "genome coverage." While "coverage" (often referred to as "depth of coverage") usually describes the average number of reads per base, "breadth of coverage" refers to the proportion of the genome covered by at least one read or a minimum read threshold. These concepts are not clearly distinguished in the manuscript, which could lead to confusion.

Line 56: The phrase "associations" is vague. Could you specify what types of associations are being referenced (e.g., between genome coverage and sample metadata, environmental variables, etc.)?

Line 125: Species names should be italicized. Please check the entire manuscript for proper taxonomic formatting and other typographical errors.

Line 134: The term "position of coverage" is unclear. Please clarify what is meant—do you refer to specific genomic coordinates with read support, or something else?

Line 245: The correct term is likely "cumulative."

Lines 321–322: The reference formatting is incorrect here, as it is in several other parts of the manuscript. A thorough check of the references section is needed.

Line 363: The phrase "other nice regions" is vague and informal. Please provide a more precise description of these regions and why they are noteworthy.

Reviewer #2

(Remarks to the Author)

In this manuscript Weng et. al. present a new program named micov. This program computes position-specific coverage across microbial genomes and synthesizes the information using statistical analyses and graphs. The authors then apply this program to a number of different datasets to showcase its utility.

The manuscript does a great job of applying the tool to a wide range of datasets while keeping the total length of the manuscript relatively short (kudos for this!). However the manuscript is missing data to back up some of the claims, it's unclear about how some core analyses were performed, and the overall utility of the tool as compared to existing programs is not entirely clear. That said, the documentation and source-code available on GitHub are top-notch, which greatly increases the accessibility of the program to other users.

Sincerely,
Dr. Matthew Olm

Specific comments:

- 1) Throughout the manuscript, including in the first sentence of the main text, the authors use "genome coverage" to refer to the fraction of a reference genome covered by at least one read. This is a little confusing and different from the mainstream—many people use the term "genome coverage" to refer to the average number of reads mapping to any base in the genome (e.g. a genome at 8x coverage has, on average, 8 reads mapping to each position). The terms "genome breadth" or "genome detection" are often used to refer to the fraction of a reference genome covered by at least one read. It's not a requirement to change the terminology in the paper, as this paper makes it clear how it's using the terms it does, but I personally believe changing "coverage" to "breadth" would improve readability.
- 2) I struggle to interpret the "cumulative coverage" plots that make up the core of this program. Adding a paragraph to the main text describing how to interpret these plots would be very welcome. For example, I do not understand how the claim "Comparing cumulative coverages among groups suggested detection of *P. copri* in samples where the region was absent was not a false positive (Figure 2b)" (line 92) is arrived at by looking at Figure 2b. It also seems to be that these plots assume that everyone in each population has the same organism present, which may be a bold assumption? Fundamentally this just seems like an odd way of plotting the data, and I would like to know the reasoning behind plotting the data in this way and specifically how to interpret these plots (as they seem to be the major output of micov).
- 3) The program says that it offers "two key advances: rapid sample type-specific cumulative coverage calculations and differential coverage detection along genomes." For the first claim, it would be great to profile the speed of micov compared to existing tools (e.g. coverM); there's currently no data to support the claim that micov is "rapid". For the second claim, I don't see any instructions on GitHub about how to complete this statistical analysis. If this is claimed as a major advance of the program, instructions on how to perform the analysis without re-implementing it yourself based on the methods section should be provided.
- 4) It's unclear to me how regions like PC351 and L682 were originally identified. Could you please clarify this, and also please explicitly state if FDR correction was used in the identification?
- 5) On line 138-141 it's stated that Anvi'o "visualizes average coverages" while micov "is uniquely designed to visualize and analyze coverage variation at a finer genomic resolution". This doesn't seem entirely correct, as I believe a core functionality of Anvi'o is the ability to visualize fine-grained coverage patterns of a genome across several samples simultaneously.
- 6) Figure 1 was not very informative for me personally, I feel that using this figure to showcase how the statistical analyses are performed and/or how to interpret the cumulative coverage plots would be more useful. I also do not understand why the legend says "Bottom left: Non-cumulative plots are not shown."
- 7) Figure 2 has many distracting light grey lines rendered on it, and the order of the x-axis groups in Figure 2a is odd (why is Mexico (present) not with the other "present" groups?).
- 8) For the analysis presented in Figure 2h, my interpretation of the data is that many sequencing runs have to be performed, aggregated, and combined for each group in order to detect EPEC. Because all group lines start at "0", that implies that each group misses EPEC at least some of the time. It also shows that EPEC is very often "detected" when it's not present at all (since over half of the blue line rises to 100%). Are these interpretations correct? If so, it's unclear why this would be an effective strategy to identify EPEC in wastewater.
- 9) In plot 2h, it seems an unfair comparison to have so many more samples of "0" than all other groups. It seems that presenting "percentile" rather than "sample number" on the x-axis of this plot would be a good way of correcting this bias.
- 10) For the analysis presented in Figure 2j, could you please compare these results to what you would get with a simple "presence / absence" based statistical test?

Reviewer #3

(Remarks to the Author)

The authors present a tool called micov that compares the proportion of genomes covered by sequencing reads across samples. The tool can also highlight differential coverage of particular regions of a genome. This information can be used to identify interesting taxa or genomic regions. For instance, the authors discovered a key genomic region of *P. copri* that explains more variation than country in a gut microbiome dataset.

While the algorithm appears to be well defined and the tool fills a useful niche, I have some concerns about the text and running the tool below.

1. Coverage is typically used to mean the average read coverage across a genome/contig (e.g. in CoverM and Anvi'o), rather than the proportion of the genome covered by at least one read. I think it would be clearer if the Authors used covered fraction or a similar phrase rather than coverage when ambiguous.
 2. Is there a cutoff for coverage? Or is a single read sufficient? Line 290 indicates that a single read is sufficient. Is that an appropriate cutoff for presence/absence?
 3. It is unclear the purpose of using cumulative coverage. If I understand correctly, it refers to the total proportion of the genome covered by reads across different samples. But then what is being accumulated? If a section has coverage in an earlier sample but not the current one, it is included in the proportion covered? Lines 241-245 are ambiguous, particularly line 244: "and the lower ranked samples". And the presence of "non-cumulative coverage plots" (line 260-263). While these plots could be useful in comparing samples grouped by metadata, the meaning is unclear when the groups are defined by coverage of a particular region, like in Figure 2b.
 4. More specifically, what is the advantage of the cumulative coverage plot and statistics over analysis of a bar plot or coverage distribution plot.
 5. What "variation" was used to select Lachnospiraceae? I would prefer a multiple testing correction if the "variation" is variation in coverage across plant consumption, which is the value that you are then testing. Since there is no a priori reason for selection, this may be cherry-picking.
- I also tried to run micov on my own samples. The pipeline did not complete successfully due to the following issues.
6. Only some `cov.gz` files were generated. Most were empty, with no log or error to indicate what went wrong. The `.sam` files are not empty.
 7. The README references the command `per-sample-group`, which is not implemented. I expect this is supposed to be `per-sample`.
 8. While the parquet files were created successfully, `per-sample` with `--parquet-coverage`, `--plot`, `--sample-metadata`, `--sample-metadata-column`, `--features-to-keep` and `--output` produced no files, without any logging or errors.

Minor Comments

Figure 2b: Inset text is too small.

Figure 2h: Inset text is too small. The cumulative lines indicate that some samples have 0% coverage of the spike-in. Is that expected? What proportion had <5% coverage?

Figure 2i: Legend location obscures green line. Also inset text is too small.

Version 1:

Reviewer comments:

Reviewer #1

(Remarks to the Author)

The authors have addressed the suggestions and incorporated the requested modifications.

I believe the manuscript can now be accepted for publication in its current form.

Sincerely

Reviewer #2

(Remarks to the Author)

Authors have adequately addressed all of my previous comments, and I now believe the manuscript is suitable for publication in Communications Biology.

Reviewer #3

(Remarks to the Author)

I appreciate the improved explanation of cumulative coverage. There is a lot of supporting analysis present only in the response to reviewers document. Are these going to be added to a supplemental document? I found the analysis in the response to Reviewer 2 point 10 particularly useful.

Most of my concerns have been adequately addressed, however there are a few that remain.

1. Do you also have benchmarking comparison for thread counts >1? Researchers are much more likely to use HPC systems for this type of analysis, so it would be useful to compare across a range of thread counts.
2. It is unclear why Figure R2 is not used in place of Figure 2H. From the response text, the difference appears to be the removal of water controls for the spike ins. Why these were included in the analysis is not explained. With their addition, Figure 2H does not represent realistic sample groups, and is therefore of little use for practical interpretation. Additionally,

you state that it only includes samples with non-zero coverage. It would be helpful to have the number of samples with non-zero coverage stated in the legend, to give an idea of how many samples were necessary to get detection of the claimed "one genomic copy".

Previously, I installed micov using its conda version (2025.2), but since that hasn't been updated, I switched to installing via pip. I have now run the new version of micov using my own samples. Thank you for fixing the bugs. I especially appreciate that fixing the bug caused by the click library must have been frustrating.

Reviewer #1 (Remarks to the Author):

General comments:

In the paper “Calculating fast differential genome coverages among metagenomic sources using micov”, Yuhan Weng and colleagues present a tool designed to compute and compare per-sample genome coverage positions across multiple genomes and samples.

While the concept behind the tool is not entirely novel, micov could prove useful when working with large-scale datasets involving numerous samples.

However, the paper is currently challenging to follow due to complex sentence structures and would benefit from a substantial reorganization, especially in the “Main” section. Simplifying the language would significantly improve readability. Additionally, there are several grammatical mistakes and formatting issues (e.g., in the references) that need to be addressed.

Author response: We appreciate this valuable feedback. As suggested by the reviewer, we have thoroughly revised the manuscript to improve clarity and flow, simplifying complex sentence structures, especially in the “Main” section. We also performed a comprehensive grammar and formatting check, including reference formatting, to address all issues. Changes are reflected throughout the revised manuscript, particularly in the "Main" and "References" sections.

One of the more promising features of micov is its potential to investigate genomic variation across different samples for one or more species. Nevertheless, there is not a comparison with other tools performing strain characterization in metagenomic samples.

Author response: We thank the reviewer for this suggestion and have now added a dedicated comparison paragraph from lines 187 to 197 in the revised manuscript:

Although existing coverage calculation tools like CoverM¹ and Anvi'o² support coverage computation and visualization, micov outputs a table ranking genomic regions by decreasing coverage variation across sample groups. This functionality can highlight mobile elements, or other strain variations, that are specific to a subgroup of samples. micov also supports the computation of coverages at least as fast as the current state-of-the-art tool CoverM (Supplementary Figure S2, Supplementary Table 3). micov's cumulative coverage calculations represent a new feature that, to the best of our knowledge, is not available in other tools. Although widely used tools such as InStrain³, MIDAS2⁴, and StrainPhlAn^{4,5,6}, excel at detecting single-nucleotide variations for high-resolution strain profiling, micov employs a coverage-based approach that complements existing SNP-focused strain characterization tools to identify genomic segments with differential coverage patterns between sample groups.

Also, the main target and intended use of the tool should be explained more clearly at the beginning—this only becomes apparent after reading a large portion of the manuscript. Clarifying these points and improving overall readability would significantly strengthen the paper.

Author response: We think this is an excellent suggestion. We have clarified the primary goals and intended applications of micov in the beginning of the “Main” section (lines 63–70), emphasizing its utility for the following three applications:

First, we identified a genomic region in P. copri with a greater and independent effect on overall microbiome composition than the known large effect of country of origin. Second, we uncovered a novel dietary association with a partially annotated region in an uncharacterized Lachnospiraceae taxon, which enabled hypothesis generation for genes with unknown functions in an annotation-independent manner. Finally, we showed that we can detect a single genomic copy of EPEC in wastewater, and also differentiate the presence of M. gnavus in distinct types of low biomass specimens.

Specific Comments:

Line 46: Consider referring to “breadth of coverage” instead of just “genome coverage.” While “coverage” (often referred to as “depth of coverage”) usually describes the average number of reads per base, “breadth of coverage” refers to the proportion of the genome covered by at least one read or a minimum read threshold. These concepts are not clearly distinguished in the manuscript, which could lead to confusion.

Author response: We thank the reviewer for this suggestion. We agree that the term ‘genome coverage’ could be ambiguous and that ‘breadth of coverage’ potentially provides clarity here. As suggested by the reviewer, we now use the term “breadth of coverage” at the beginning of the “Main” section (line 46). In addition, we have explicitly stated in the “Main” section that “*Throughout this article, we use ‘coverage’ to refer specifically to breadth of coverage (fraction of reference genome covered by at least one read), not depth of coverage (average reads per base), unless otherwise specified*” (lines 74-76). We have also updated the terms used in Figure 1 (line 215). We hope these revisions help clarify the key concepts early on in the manuscript and improve the manuscript’s readability.

Line 56: The phrase “associations” is vague. Could you specify what types of associations are being referenced (e.g., between genome coverage and sample metadata, environmental variables, etc.)?

Author response: We agree with the reviewer’s assessment. Accordingly, we revised the text to clarify that the associations refer to associations between

breadth of coverage patterns and sample metadata including phenotypic traits and environmental variables (lines 60-61):

These comparative analyses reveal associations between breadth of coverage patterns and sample metadata, including phenotypic traits and environmental variables, that cannot be seen when all samples are combined.

We thank the reviewer for helping us improve the clarity of the manuscript.

Line 125: Species names should be italicized. Please check the entire manuscript for proper taxonomic formatting and other typographical errors.

Author response: We have corrected all taxonomic formatting, ensuring species names are italicized throughout the manuscript.

Line 134: The term “position of coverage” is unclear. Please clarify what is meant—do you refer to specific genomic coordinates with read support, or something else?

Author response: We thank the reviewer for pointing this out. The reviewer is correct that “positions of coverage” here refers to specific genomic coordinates with read support. This phrasing serves as a transition from the cumulative curve analysis to our second visualization approach, which displays the breadth of coverage patterns across the genome from start to finish (not cumulative). To improve clarity, we have revised the text (lines 182-183) to better explain this transition.

Line 245: The correct term is likely “cumulative.”

Author response: We thank the reviewer for this suggestion and have corrected the term to “cumulative coverage” (line 415).

Lines 321–322: The reference formatting is incorrect here, as it is in several other parts of the manuscript. A thorough check of the references section is needed.

Author response: We thank the reviewer for pointing this out. We have corrected the abovementioned reference formatting (lines 620-621) and updated the reference formatting throughout the manuscript.

Line 363: The phrase “other nice regions” is vague and informal. Please provide a more precise description of these regions and why they are noteworthy.

Author response: We thank the reviewer for noting this informal phrasing. We have revised the text to replace “*other nice regions*” with “*other top-ranked regions*”

identified by binning (Methods); their associated PERMANOVA statistics can be found in Supplementary Table 2 (sheet: thdmi-wgs-weighted-raw)" (lines 538-540).

The top-ranked regions are noteworthy because they exhibit the highest variation in coverage patterns across metadata groups. As detailed in the Methods section (lines 453-461), the binning approach divides each reference genome into N non-overlapping bins and ranks these bins based on the standard deviation of sample hits across sample groups. Bins with greater variability across groups are ranked higher.

In the downstream PERMANOVA analysis, we focused on these highly variable regions to assess their effect sizes on the overall microbiome composition. While we highlighted the PC351 region in the main text due to its particularly strong effect (exceeding that of geographic country in the THDMI dataset), we have included the full set of PERMANOVA statistics for the other top-ranked regions in Supplementary Table 2 (sheet: thdmi-wgs-weighted-raw) for completeness.

We appreciate the reviewer's comment, which helped us improve the precision and clarity of the manuscript.

Reviewer #2 (Remarks to the Author):

In this manuscript Weng et. al. present a new program named micov. This program computes position-specific coverage across microbial genomes and synthesizes the information using statistical analyses and graphs. The authors then apply this program to a number of different datasets to showcase its utility.

The manuscript does a great job of applying the tool to a wide range of datasets while keeping the total length of the manuscript relatively short (kudos for this!).

Author response: We thank the reviewer for this comment.

However the manuscript is missing data to back up some of the claims, it's unclear about how some core analyses were performed, and the overall utility of the tool as compared to existing programs is not entirely clear.

Author response: We appreciate this valuable feedback from the reviewer and have made the following improvements to clarify our approach for core analyses and how it compares with existing tools.

First, we have added further explanation of micov's major visualizations and associated statistical analysis in the revised manuscript (lines 80-116). The concept of cumulative coverage breadth curves are further discussed in detail below (Reviewer #2, specific comments 2 and 4).

Secondly, We have added runtime comparisons to illustrate micov's performance in Supplementary Figure S2 and we have also added a paragraph to the manuscript comparing micov with other tools in lines 187-197 in the revised manuscript.

We appreciate the use for asking these important questions and for helping us improve the manuscript.

That said, the documentation and source-code available on GitHub are top-notch, which greatly increases the accessibility of the program to other users.

Author response: We appreciate the reviewer taking the time to review our repository.

Sincerely,
Dr. Matthew Olm

Specific comments:

1) Throughout the manuscript, including in the first sentence of the main text, the authors use "genome coverage" to refer to the fraction of a reference genome covered by at least one read. This is a little confusing and different from the mainstream- many people use the term "genome coverage" to refer to the average number of reads mapping to any base in the genome (e.g. a genome at 8x coverage has, on average, 8 reads mapping to each position). The terms "genome breadth" or "genome detection" are often used to refer to the fraction of a reference genome covered by at least one read. It's not a requirement to change the terminology in the paper, as this paper makes it clear how it's using the terms it does, but I personally believe changing "coverage" to "breadth" would improve readability.

Author response: We thank the reviewer for this suggestion. We agree that the term 'genome coverage' is different from the mainstream and could lead to confusion. As recommended, we now use the term "breadth of coverage" at the beginning of the "Main" section (line 46). In addition, we have explicitly stated in the "Main" section that "*Throughout this article, we use 'coverage' to refer specifically to breadth of coverage (fraction of reference genome covered by at least one read), not depth of coverage (average reads per base), unless otherwise specified*" (lines 74-76). We have also updated the terms used in Figure 1 (line 215). We hope these revisions help clarify the key concepts early on in the manuscript and improve the manuscript's readability.

2a) I struggle to interpret the “cumulative coverage” plots that make up the core of this program. Adding a paragraph to the main text describing how to interpret these plots would be very welcome.

Author response: We thank the reviewer for this valuable feedback and for highlighting the need for clearer explanation. Before addressing the specific comments in here, we would like to offer additional background on the derivation of the plots. The idea for cumulative coverage came from multiple exposure photography in astronomy. In any given photograph, the number of photons of light from a faint object is impossible to distinguish from background noise. However, repeated observations of the same object accumulate enough signal in the same place to distinguish the object from background noise. Extending this idea to metagenomics, we reasoned that it might be impossible to distinguish a signal stemming from a small number of sequences matching a genome in one sample from background noise, but that if the genome were truly present then accumulation of the genome across many samples should yield a pattern of random accumulation over the full length of the genome. Furthermore, if the genome were present in some categories of samples but not others (for example, present in stool from individuals with a specific disease but absent from individuals without that disease), then the accumulation curves would differ in sets of samples grouped according to these categories.

Accordingly, our first use of cumulative coverage was an unpublished application for gathering evidence of the presence of a genome similar to a reference from a large collection of low biomass metagenomic samples, stratified by clinical metadata. The intuition was that we can reasonably assume an individual sample will have a handful of reads approximately uniformly distributed over the genome if it is present but the small counts may not be distinguishable from background. If coverage increases randomly across the genome when adding samples, especially if coverage only increases with samples within a specific category, this adds support to the hypothesis that this particular genome or a related one is present in the sample. This pattern would also rule out alternative hypotheses such that the matches are driven by lateral transfer to another organism or convergent evolution (accumulation would saturate at coverage of only a small part of the genome), contamination (accumulation would be unrelated to sample category), etc.

In support of the reviewers comment, we have added text on lines 82-109 for providing additional interpretation of the cumulative coverage plots.

2b). For example, I do not understand how the claim “Comparing cumulative coverages among groups suggested detection of *P. copri* in samples where the region was absent was not a false positive (Figure 2b)” (line 92) is arrived at by looking at Figure 2b.

Regarding the specific statement on 'false positive', we agree with the reviewer that Figure 2b alone does not directly demonstrate that *P. copri* detection in samples lacking PC351 is not a false positive. We have revised the text (line 133-137) to more conservatively state that

“Comparing cumulative coverage across groups, we observed a continued increase in genome breadth even in samples lacking the PC351 region (Figure 2b). Additionally, the coverage patterns along the genome were qualitatively similar between groups with and without PC351 (Figure 2a), hinting at the possibility that P. copri detection in samples lacking PC35 may not be a false positive (Figure 2b).”

2c). It also seems to be that these plots assume that everyone in each population has the same organism present, which may be a bold assumption? Fundamentally this just seems like an odd way of plotting the data, and I would like to know the reasoning behind plotting the data in this way and specifically how to interpret these plots (as they seem to be the major output of micov).

Lastly, we agree that assuming all individuals in a population have the exact same organism would be a bold and inaccurate assumption. To clarify, micov does not assume the exact reference genome is present in every sample. As with all reference-based approaches, we acknowledge that reference is an approximation. Essentially all references are wrong and we cannot conclude that the EXACT genome is in any sample even with 100% coverage as the true genome may have an insertion not represented by the reference. We are instead in the domain of population variation relative to a fixed reference. We aim to characterize population-level variation relative to a fixed reference and to identify consistent genomic regions that show differential coverage patterns across sample groups.

We appreciate the reviewer's thoughtful comments, which helped us clarify both the method and its interpretation for readers.

3) The program says that it offers “two key advances: rapid sample type-specific cumulative coverage calculations and differential coverage detection along genomes.” For the first claim, it would be great to profile the speed of micov compared to existing tools (e.g. coverM); there's currently no data to support the claim that micov is “rapid”.

Author response: We thank the review for highlighting the need to support our performance claim. We chose to focus our comparison on CoverM because its documentation explicitly states that it is a “fast DNA read coverage and relative abundance calculator focused on metagenomics applications”. Although CoverM does not compute cumulative coverage breadth, its functionality can be constrained to compute per-sample, per-genome coverage breadth by using the following command: ``coverm genome --bam-files -m covered_fraction``. The covered fraction

is a metric that micov also provides, making it a relevant and practical benchmark for evaluating runtime performance.

To evaluate performance, we benchmarked micov and CoverM across datasets of increasing size (from 100 to 10 million read alignments) randomly selected from the THDMI dataset to evaluate their runtime performance for coverage breadth calculation. All tests were performed using a single thread (*POLARS_MAX_THREADS=1* for micov, *-t 1* for CoverM) to avoid program-specific parallel overhead.

CoverM requires sorted BAM files prior to coverage computation, while micov is agnostic to alignment sort order. To account for this difference, we included BAM sorting time as part of CoverM's total runtime cost. In contrast, micov runs directly on unsorted SAM files, simplifying the process.

Across all input sizes, micov completed faster than CoverM when sorting time was included. For example, at 10 million alignments, micov completed in 50.3 seconds, while CoverM (including sorting) took 129.3 seconds - a 61% speedup (Supplementary Figure S2).

The coverage calculation step alone (excluding sorting) was faster for CoverM at the largest scale (40.2s v.s. 50.3s, Supplementary Table 3). We observed a sharp increase in micov's execution time from 1 million to 10 million alignments, indicating potential for improving scalability. However despite this increase, micov completes the coverage calculations at a speed comparable to CoverM, with CoverM finishing approximately 10 seconds faster and micov outperformed CoverM for input sizes up to 1 million alignments.

In support of the reviewer's comment, we added benchmarking methods (lines 627-640) and results (Supplementary Figure S2, lines 700-717) to the revised manuscript. These results reasonably support a claim that micov is rapid, as the state-of-the-art tool CoverM describes itself as fast, and micov matches or exceeds its performance.

Supplementary Figure S2 (lines 700-717): Execution time of micov and CoverM for samples of 100 to 10 million alignments.

For the second claim, I don't see any instructions on GitHub about how to complete this statistical analysis. If this is claimed as a major advance of the program, instructions on how to perform the analysis without re-implementing it yourself based on the methods section should be provided.

Author response: We thank the reviewer for their constructive comments and the help in improving the usability of this repository. In response to this comment, we have updated the GitHub repository to include step-by-step instructions for performing statistical analysis described in the manuscript. Specifically, we have

1. Added a new paragraph under section 'Generate Per-Sample-Group Plots' of the README to explain the statistical analysis automatically performed when generating cumulative coverage plots:

"Pairwise Kolmogorov-Smirnov (KS) tests between all sample groups' cumulative coverage curves are automatically conducted and results saved in cumulative.ks.tsv. The KS test quantifies whether two sample groups differ in the distribution of their cumulative genome coverages, with the KS statistic measuring the maximal difference between the two cumulative distributions, and the KS p-value assessing the statistical significance of the difference."

2. Added a new section in the README titled "Binning and Ranking" that outlines the approach used to identify genomic regions with variation in coverage patterns

across sample groups. We also included descriptions of the expected output files to help users interpret the results more easily.

“The binning command allows you to divide genome positions into fix-sized bins and compute summary statistics across samples, based on sample metadata. This is useful for identifying regions of interest (e.g. high variability across samples).

...

Each bin is ranked based on the standard deviation of sample hits across groups associated with the chosen metadata category, with bins exhibiting higher variability ranked at the top.

The rankings are saved in the output stats_by_variance_of_sample_hits.tsv whereas binning statistics (start and end positions of each bin, number of sample hits per bin, number of read hits per bin.etc) are saved in stats_bins.tsv.”

3. Provided the following example that outlines the sample command for the abovementioned analysis.

...

```
micov binning \  
  --parquet-coverage ./example/parquet/example \  
  --sample-metadata ./example/metadata/sample_metadata.txt \  
  --features-to-keep ./example/metadata/feature_metadata.txt \  
  --metadata-variable "dog" \  
  --outdir ./example/binning \  
  --rank
```

...

The updated instructions, along with the complete test command can be accessed at <https://github.com/biocore/micov/tree/main>. We thank the reviewer again for helping us improve the accessibility of the repository and believe that these additions make it easier for users to perform the statistical analysis without needing to re-implement any part of it from scratch.

4) It's unclear to me how regions like PC351 and L682 were originally identified. Could you please clarify this, and also please explicitly state if FDR correction was used in the identification?

Author response: We thank the reviewer for this comment. PC351 and L672 were identified using the binning approach outlined in line 453-461 under the Methods section.

Specifically, for PC351, the reference genome *P. copri* was first divided into non-overlapping bins and a sample is considered present in the bin if at least one read

from the sample is within the bins corresponding start and stop coordinates. The standard deviation of these counts (“sample hits”), across sample groups, is computed. The resulting output is ranked by the standard deviation. We evaluated the top 10 ranked regions for *P. copri* differentiating country and picked PC351 as it had the largest observed effect size as measured by PERMANOVA. The effect sizes of other regions from the top 10 can be found in Supplementary Table 2 (sheet: thdmi-wgs-weighted-raw).

For L682, we applied the same binning method to the reference genome *Lachnospiraceae* (GCA_900066105.1, G900066105 in Web of Life version 2). This genome was selected *a priori* given the prior evidence linking Lachnospiraceae to dietary plant consumption in the broader project these samples are part of. Specifically, multiple studies based on the American Gut Project (AGP) have consistently identified *Lachnospiraceae* as enriched in individuals with plant-rich or fermented plant-based diets. This includes associations with high plant diversity (McDonald & Hyde *et al.*, 2018), fermented plant consumption (Taylor *et al.*, 2020), and plant-based dietary patterns compared to Western and other diets (Cotillard *et al.*, 2022).

Given this consistent link to plant-rich diets, we selected this genome to assess whether any specific regions, like L682, were particularly associated with dietary patterns. We thank the reviewer for raising this important question and have expanded the rationale for selecting PC351 and L682 in the revised manuscript (lines 494-501 and 506-514, respectively).

5) On line 138-141 it's stated that Anvi'o "visualizes average coverages" while micov "is uniquely designed to visualize and analyze coverage variation at a finer genomic resolution". This doesn't seem entirely correct, as I believe a core functionality of Anvi'o is the ability to visualize fine-grained coverage patterns of a genome across several samples simultaneously.

Author response: Thank you for this important distinction. In response to this comment, we ran a subset of THDMI samples using Anvi'o and confirmed that it produces a fine-grained genomic resolution of coverage across the genome (**Figure R1**).

Figure R1: Anvi'o visualization with fine-grained genomic resolution for a subset of THDMI data ($n = 30$).

It is correct that Anvi'o is capable of a fine-grained and interactive genomic resolution of coverage across the genome.

Although Anvi'o enables coverage visualization across samples, it lacks built-in functionality for highlighting genomic regions with the high variation in coverage patterns across samples. In contrast, micov outputs a table ranking genomic regions by decreasing coverage variation across sample groups (See 'Binning' section in Methods). This functionality could potentially highlight mobile elements or other strain variations, that are specific to a subgroup of samples.

We have updated line 187-193 with the following text:

Although existing coverage calculation tools like CoverM¹ and Anvi'o² support coverage computation and visualization, micov outputs a table ranking genomic regions by decreasing coverage variation across sample groups. This functionality can highlight mobile elements, or other strain variations, that are specific to a subgroup of samples. ... micov's cumulative coverage calculations represent a new feature that, to the best of our knowledge, is not available in other tools.

We thank the reviewer for this important clarification and for helping improve the clarity of the manuscript.

6) Figure 1 was not very informative for me personally, I feel that using this figure to showcase how the statistical analyses are performed and/or how to interpret the cumulative coverage plots would be more useful. I also do not understand why the legend says “Bottom left: Non-cumulative plots are not shown.”

Author response: We thank the reviewer for this feedback. Although figure 1 currently provides a schematic of micov’s workflow, we agree with the reviewer’s suggestion that this figure could benefit from more details regarding the statistical analyses being performed and the interpretation for main visualizations produced. We have updated Figure 1 (lines 215-227) to incorporate these elements. Specifically,

1. To improve the clarity of the schematic, in the ‘Cumulative Coverage’ panel, we changed the y-axis label from ‘coverage’ to ‘cumulative coverage breadth’. In both ‘Cumulative Coverage’ and ‘Scaled/Unscaled Position Plots’, we changed the x-axis label to ‘samples ordered by coverage breadth’ to help readers understand the ordering of samples along the x-axis.

2. To help with interpretation of the main visualizations, we added color legends in the ‘Cumulative Coverage’ panel, illustrating that cumulative coverages from different sample groups are plotted in the same figure. Additionally, we updated the figure legend with the following explanatory text: “*This display helps assess whether coverage continues to accumulate across a sample group, or instead plateaus early to suggest that only a small part of the genome is represented in the sample. Position plots display coverage patterns across a reference genome.*” (lines 218-221).

3. We added two panels corresponding to the statistical analyses performed by micov following the generation of each visualization, offering a preview of expected outputs. After cumulative coverage plots are generated, a Kolmogorov-Smirnov (KS) test is conducted to quantify differences between chosen sample groups, especially when visual inspection is challenging due to overlapping curves. Key columns of the output include sample groups being compared, KS statistic, and p -value. For genomic region variation, the genome is divided into N bins, and regions are ranked based on the standard deviation of sample hits across groups. Key columns of the output include the ranking, genome id, start and end position of the genomic region, and standard deviation of sample hits. We have expanded the figure legend to guide the interpretation of Figure 1 (lines 221-227).

Finally, we thank the reviewer for drawing attention to the confusing statement in the figure legend. To simplify the legend, and given the misunderstanding, we have removed the statement on non-cumulative plots.

Figure 1 (lines 215-227): Schematic of the micov workflow. Cumulative coverage represents the total coverage breadth achieved when samples are ordered by increasing individual coverage breadth and added sequentially. This display helps assess whether coverage continues to accumulate across a sample group, or instead plateaus early to suggest that only a small part of the genome is represented in the sample. Position plots display coverage patterns across a reference genome. After cumulative coverage plots are generated, a Kolmogorov-Smirnov (KS) test is conducted to quantify differences between chosen sample groups, especially when visual inspection is challenging due to overlapping curves. Key columns of the output include sample groups being compared, KS statistic, and p-value. For genomic region variation, the genome is divided into N bins, and regions are ranked based on the standard deviation (std) of sample hits across

groups. Key columns of the output include the ranking, genome id, start and end position of the genomic region, and standard deviation of sample hits (Methods).

7) Figure 2 has many distracting light grey lines rendered on it, and the order of the x-axis groups in Figure 2a is odd (why is Mexico (present) not with the other “present” groups?).

Author response: We thank the reviewer for this invaluable insight. In Figure 2, the sample groups are ordered by increasing sample size, which we have now clarified in the figure legend (line 235). From left to right, the UK (present) has the fewest samples, followed by the US (present), and so on. The grey lines we believe the reviewer is referring to are gridlines and simply a visual aesthetic to help a reader understand the data relationship to the genome coordinates on the y-axis. Similarly, we have added clarification to the figure legend (lines 235-237).

8) For the analysis presented in Figure 2h, my interpretation of the data is that many sequencing runs have to be performed, aggregated, and combined for each group in order to detect EPEC. Because all group lines start at “0”, that implies that each group misses EPEC at least some of the time. It also shows that EPEC is very often “detected” when it’s not present at all (since over half of the blue line rises to 100%). Are these interpretations correct? If so, it’s unclear why this would be an effective strategy to identify EPEC in wastewater.

Author response: We thank the reviewer for raising this important question. The reviewer is correct that many sequencing runs have to be performed, aggregated, and combined for each group in order to detect EPEC.

Regarding the comment that EPEC is very often “detected” when it’s not present, we would like to further clarify the reason for this observation in this application. This occurs because wastewater naturally contains background E. coli that shares genomic similarity with EPEC, thus, wastewater represents one of most difficult types in terms of genome detection as we have to deal with background burden of the same species. Therefore, we pursued analysis of this sample-type to determine the robustness of genome detection via micov. We would also like to point out that the dataset that we used for this analysis had also contained some water control samples, we apologize that this clarification was not present in the first submission and have now added clarification from in the Methods section (lines 566-567) as well as Figure 2 legend (line 248). Water controls were present across all groups, including non spike-in and spike-in groups, and they do not possess background microbial burden. This also in-part explains the higher number of ~0 coverage samples in the zero spike-in group vs the other groups.

Therefore, to provide a clearer analysis using micov for EPEC detection, we conducted a revised analysis using only wastewater samples and excluded water

controls (**Figure R2**). Therefore, we provide this updated analysis as a more wastewater-focused assessment of this challenge.

The observed “detection” of EPEC in non-spike-in wastewater samples is expected due to the background *E. coli* present in the wastewater. Therefore, the background *E. coli* contributes to the signal in the non-spike-in samples, which is why there is detection in this group. This property of wastewater means that detecting certain pathogens will be very difficult by using more traditional read mapping strategies. As such, this application represents one of the most difficult situations in which strain detection may be pursued.

Thus, when running many negative control samples against the EPEC genome, we should expect to find that coverage will converge to some degree commensurate with the overall similarity of the background *E. coli*. We broadly observe this when considering at least 100 control wastewater samples. However, when assessing the samples with spike-ins, we observe that we can achieve coverage 'saturation' with significantly fewer samples. For the lowest spike-ins (0.1) there are 71 samples, and for the highest spike-in (1000) there are 24 samples (**Figure R2**). This demonstrates that we can address the complex problem of detecting a specific pathogen against a high background of similar species using micov.

Figure R2: Cumulative coverage figure of wastewater samples only.

The reviewer correctly noted that EPEC is sometimes missed. This is expected and may be due to variable sample extraction, library prep, and sequencing quality during standard processing.

Clarification of Figure 2h

We understand the confusion about lines starting at "0." Figure 2h displays only samples with non-zero EPEC coverage, with the lowest coverage being 0.002%. This is because the primary purpose of micov's visualization is to examine how coverage patterns vary across groups in samples with some evidence of the reference genome. We have clarified this in the updated figure legend for Figure 2h (lines 250-251). Additionally, we have updated the figure legend (line 248) and the relevant method section (lines 566-567) to clarify that Figure 2h included both wastewater and water control samples.

Figure 2h updated

In addressing reviewer comments, we have observed a minor discrepancy between versions of the metadata for the EPEC dataset. We have since caught this error and have updated figure 2h and Supplementary Table 2 (sheet: EPEC-raw) using the most up-to-date metadata (line 230). Major conclusions from the dataset remain robust.

We thank the reviewer for this insightful question, which prompted us to improve the explanation of Figure 2h and conduct a more focused analysis using wastewater-only samples.

9) In plot 2h, it seems an unfair comparison to have so many more samples of "0" than all other groups. It seems that presenting "percentile" rather than "sample number" on the x-axis of this plot would be a good way of correcting this bias.

Author response: We thank the reviewer for this thoughtful suggestion. Plotting the x-axis as percentile is indeed a way to normalize for unequal group sizes. In fact, earlier versions of micov included this as an option. At the reviewers guidance, we have restored this option and updated README in PR#39 (<https://github.com/biocore/micov/pull/39>). Users can now specify `--percentile` in the command `micov per-sample-group` to display plots with x-axis representing percentile instead of sample number. We regenerated the plots for Figure 2h with and without `--percentile` (**Figure R3**).

Figure R3: Cumulative coverage figures of the same dataset with x-axis representing percentile (left) and sample numbers (right).

We had previously removed this functionality, because percentile can be misleading in cases where the sample sizes among groups are imbalanced: groups with more samples would appear to accumulate coverage more quickly at lower percentiles simply because each percentile bin contains more samples. This could make large groups appear to have stronger signals by virtue of their size, rather than reflecting true biological or technical differences. That said, we agree that offering this as an option would enhance flexibility in some scenarios and allow for alternative visualizations where appropriate. Thank you again for helping us improve the usability of micov.

10) For the analysis presented in Figure 2j, could you please compare these results to what you would get with a simple “presence / absence” based statistical test?

Author response: We thank the reviewer for raising this thoughtful question. Figure 2j contains Kolmogorov-Smirnov test results comparing the cumulative coverage breadth of *Mediterraneibacter gnavus* among prospectively acquired, surgically resected, paired human mucosal and adipose tissue samples from patients with Crohn’s disease. As these are tissue samples, they are anticipated to be low microbial biomass.

To conduct a “presence / absence” based statistical test using the same data, we first need to determine the coverage breadth threshold which defines the presence of *M. gnavus* in the sample, which is a critical parameter for such an analysis. We first plotted the number of samples with *M. gnavus* present at different coverage breadth thresholds (**Figure R4**). At thresholds higher than 78.8%, 0 out of 307 samples have *M. gnavus* present. Therefore, we limit our analysis to thresholds below 78.8% and approached this analysis using five different coverage thresholds 0.1%, 1%, 10%, 50%, and 70%. Note that the cumulative coverage plot achieves a higher coverage than any individual biological sample, which would occur if the genome (or a near relative) is present and being observed with approximately uniformly sampled short sequencing reads.

Figure R4: Number of samples with *M. gnavus* present at different coverage thresholds.

At each threshold, we classified samples as *M. gnavus* present or absent based on the threshold and plotted their distributions across tissue types. We then perform pairwise Fisher’s Exact Tests between them. *P*-values are corrected using the Benjamini-Hochberg method to remain consistent with the approach used in 2j. Only significant comparisons are shown, ordered by increasing adjusted *p*-values.

At lower thresholds (0.1% and 1%), we observed a strong association of *M. gnavus* presence and involved mucosa compared to other tissue types including subcutaneous fat, negative controls, creeping fat, supporting results outlined in 2j. In addition, the comparison between involved mucosa and mesenteric fat is also significant (**Figure R5, R6**).

At medium thresholds (10% and 50%), significant differences remained between involved mucosa and the same tissue types, although the ranking of significant comparisons varied: at the 10% threshold, the most significant comparison was between involved mucosa and negative controls (**Figure R7**), whereas at the 50% threshold, it was observed between involved mucosa and mesenteric fat (**Figure R8**).

However, at the 70% threshold, *M. gnavus* was absent from the majority of samples, and no statistically significant associations were observed (**Figure R9**). These results highlight how the choice of coverage breadth threshold can dramatically influence outcomes in presence/absence analyses.

We respectfully note that determining an appropriate coverage threshold for presence/absence analysis using short reads is inherently challenging, and to the best of our knowledge, the state-of-the-art is to use an arbitrary value. Coverage breadth can be affected by factors such as genome abundance, genome size, and sequencing depth. In low-biomass settings, it is particularly difficult to distinguish between true low-abundance genomes and false positives, both of which often display low coverage.

The cumulative coverage approach we propose in micov and used in Figure 2i and 2j bypasses the selection of a coverage breath threshold, and considers the continuum of possible thresholds. It demonstrates sensitivity for genome detection and differentiation in low-biomass settings by leveraging the cumulative growth of coverage breadth across all samples (Figure 2i).

Figure R5: Presence/Absence of *M. gnavus* across tissue types. Coverage breadth threshold for organism presence is 0.1%.

Tissue Type		Counts A (present, absent)	Counts B (present, absent)	Odds Ratio	Raw p-value	Adjusted p-value (BH)	Significant ($\alpha=0.05$)
Involved Mucosa	Negative Controls	[27, 22]	[3, 35]	14.318182	0.000003	0.000064	True
Involved Mucosa	Subcutaneous Fat	[27, 22]	[5, 35]	8.590909	0.000044	0.000466	True
Creeping Fat	Involved Mucosa	[8, 34]	[27, 22]	0.191721	0.000520	0.003640	True
Negative Controls	Uninvolved Mucosa	[3, 35]	[20, 31]	0.132857	0.001149	0.006030	True
Involved Mucosa	Mesenteric Fat	[27, 22]	[19, 49]	3.165072	0.003998	0.016794	True
Subcutaneous Fat	Uninvolved Mucosa	[5, 35]	[20, 31]	0.221429	0.004936	0.017275	True

Figure R6: Presence/Absence of *M. gnavus* across tissue types. Coverage breadth threshold for organism presence is 1%.

Tissue Type A	Tissue Type B	Counts A (present, absent)	Counts B (present, absent)	Odds Ratio	Raw p-value	Adjusted p-value (BH)	Significant (α=0.05)
Involved Mucosa	Negative Controls	[16, 33]	[0, 38]	inf	0.000035	0.000366	True
Involved Mucosa	Subcutaneous Fat	[16, 33]	[0, 40]	inf	0.000030	0.000366	True
Involved Mucosa	Mesenteric Fat	[16, 33]	[3, 65]	10.505051	0.000063	0.000439	True
Creeping Fat	Involved Mucosa	[1, 41]	[16, 33]	0.050305	0.000236	0.001241	True
Subcutaneous Fat	Uninvolved Mucosa	[0, 40]	[10, 41]	0.000000	0.002120	0.008904	True
Negative Controls	Uninvolved Mucosa	[0, 38]	[10, 41]	0.000000	0.004241	0.014842	True
Creeping Fat	Uninvolved Mucosa	[1, 41]	[10, 41]	0.100000	0.010886	0.032657	True
Mesenteric Fat	Uninvolved Mucosa	[3, 65]	[10, 41]	0.189231	0.014890	0.039086	True

Figure R7: Presence/Absence of *M. gnavus* across tissue types. Coverage breadth threshold for organism presence is 10%.

Tissue Type		Counts A (present, absent)	Counts B (present, absent)	Odds Ratio	Raw p-value	Adjusted p-value (BH)	Significant ($\alpha=0.05$)
Involved Mucosa	Mesenteric Fat	[11, 38]	[0, 68]	inf	0.000034	0.000705	True
Creeping Fat	Involved Mucosa	[0, 42]	[11, 38]	0.0	0.000706	0.006028	True
Involved Mucosa	Subcutaneous Fat	[11, 38]	[0, 40]	inf	0.000861	0.006028	True
Involved Mucosa	Negative Controls	[11, 38]	[0, 38]	inf	0.001910	0.010030	True

Figure R8: Presence/Absence of *M. gnavus* across tissue types. Coverage breadth threshold for organism presence is 50%.

Figure R9: Presence/Absence of *M. gnavus* across tissue types. Coverage breadth threshold for organism presence is 70%.

Reviewer #3 (Remarks to the Author):

The authors present a tool called micov that compares the proportion of genomes covered by sequencing reads across samples. The tool can also highlight differential coverage of particular regions of a genome. This information can be used to identify interesting taxa or genomic regions. For instance, the authors discovered a key genomic region of *P. copri* that explains more variation than country in a gut microbiome dataset.

While the algorithm appears to be well defined and the tool fills a useful niche, I have some concerns about the text and running the tool below.

Author response: We thank the reviewer for their constructive comments and the help in guiding this technique to be a useful addition to the field.

1. Coverage is typically used to mean the average read coverage across a genome/contig (e.g. in CoverM and Anvi'o), rather than the proportion of the genome covered by at least one read. I think it would be clearer if the Authors used covered fraction or a similar phrase rather than coverage when ambiguous.

Author response: We thank the reviewer for this suggestion. We agree that the term 'genome coverage' is used differently by other softwares and could lead to confusion. We now use the term "breadth of coverage" at the beginning of the "Main" section (line 46). In addition, we have explicitly stated in the "Main" section that "*Throughout this article, we use 'coverage' to refer specifically to breadth of coverage (fraction of reference genome covered by at least one read), not depth of coverage (average reads per base), unless otherwise specified*" (lines 74-76). We have also updated the terms used in Figure 1 (line 215). We hope these revisions help clarify the key concepts early on in the manuscript and improve the manuscript's readability.

2. Is there a cutoff for coverage? Or is a single read sufficient? Line 290 indicates that a single read is sufficient. Is that an appropriate cutoff for presence/absence?

Author response: We thank the reviewer for raising this valuable question. Instead of assessing presence/absence of the entire reference genome, we are assessing the presence of specific genomic regions (independent of annotation) that may be shared across multiple strains or acquired through horizontal gene transfer. Since we are not making claims about complete genome presence or strain identity, but rather about the distribution of observed sequencing data within the microbial community, there is no cutoff established. Even low-coverage detection could potentially provide meaningful biological information about presence/absence patterns.

3. It is unclear the purpose of using cumulative coverage. If I understand correctly, it refers to the total proportion of the genome covered by reads across different samples. But then what is being accumulated? If a section has coverage in an earlier sample but not the current one, it is included in the proportion covered? Lines 241-245 are ambiguous, particularly line 244: “and the lower ranked samples”. And the presence of “non-cumulative coverage plots” (line 260-263). While these plots could be useful in comparing samples grouped by metadata, the meaning is unclear when the groups are defined by coverage of a particular region, like in Figure 2b.

Author response: We thank the reviewer for asking this important question. The total proportion of the genome covered by reads is being accumulated over a group of samples. Sample groupings are determined by the user based on sample metadata. The reviewer is correct that if a section has coverage in an earlier sample but not the current one, it is included in the proportion covered.

The idea for cumulative coverage came from multiple exposure photography in astronomy. In any given photograph, the number of photons of light from a faint object is impossible to distinguish from background noise. However, repeated observations of the same object accumulate enough signal in the same place to distinguish the object from background noise. Extending this idea to metagenomics, we reasoned that it might be impossible to distinguish a signal stemming from a small number of sequences matching a genome in one sample from background noise, but that if the genome were truly present then accumulation of the genome across many samples should yield a pattern of random accumulation over the full length of the genome. Furthermore, if the genome were present in some categories of samples but not others (for example, present in stool from individuals with a specific disease but absent from individuals without that disease), then the accumulation curves would differ in sets of samples grouped according to these categories.

Accordingly, our first use of cumulative coverage was an unpublished application for gathering evidence of the presence of a genome similar to a reference from a large collection of low biomass metagenomic samples, stratified by clinical metadata. The intuition was that we can reasonably assume an individual sample will have a handful of reads approximately uniformly distributed over the genome if it is present but the small counts may not be distinguishable from background. If coverage increases randomly across the genome when adding samples, especially if coverage only increases with samples within a specific category, this adds support to the hypothesis that this particular genome or a related one is present in the sample. This pattern would also rule out alternative hypotheses such that the matches are driven by lateral transfer to another organism or convergent evolution (accumulation would saturate at coverage of only a small part of the genome), contamination (accumulation would be unrelated to sample category), etc.

To help clarify uses of cumulative coverage, we have added the above text to the revised manuscript (lines 82-109).

The “lower ranked samples” refers to all samples that have a lower coverage breadth than the current sample. Along the x axis of the cumulative coverage figure, samples are ranked from left to right in increasing order of genome coverage breadth for the reference genome of interest. So at any point on the x axis, the corresponding coverage breadth value refers to the accumulated coverage breadth of this sample and all samples to the left of it. We have modified text accordingly (lines 413-416):

“For each sample group, a curve is plotted, showing the total amount of coverage for that sample and all other samples with equal or lower coverage breadth, or, in other words, it represents the cumulative coverage.”

Given the common confusion among reviewers regarding “Non-cumulative coverage plots”, we have removed non-cumulative coverage plots. We thank the reviewer for helping us improve the clarity and overall readability of the manuscript.

4. More specifically, what is the advantage of the cumulative coverage plot and statistics over analysis of a bar plot or coverage distribution plot.

Author response: We thank the reviewer for this question and we have expanded our discussion on these items in the response to (3).

5. What “variation” was used to select Lachnospiraceae? I would prefer a multiple testing correction if the “variation” is variation in coverage across plant consumption, which is the value that you are then testing. Since there is no a priori reason for selection, this may be cherry-picking.

Author response: We thank the reviewer for raising this important question regarding the selection criteria for *Lachnospiraceae* for further analysis. *Lachnospiraceae* (GCA_900066105.1, G900066105 in Web of Life version 2) was not chosen based on coverage variation within our dataset. Instead, it was selected based on consistent findings from multiple prior studies from the American Gut Project (AGP) reporting its strong association with dietary plant intake.

Specifically, *Lachnospiraceae* exhibited differential abundance between individuals consuming fewer than 10 versus more than 30 plant types per week in the original AGP publication⁷ (Figure 5E, attached below for convenience)

McDonald & Hyde *et al.*, 2018 Figure 5E. Differential abundances of sOTUs (showing the most specific taxon name per sOTU) between those who eat fewer than 10 plants per week and those who eat over 30 per week.

In another study by Taylor *et al.*, which analyzed a subset of the AGP based on self-reported consumption of fermented food, in particular fermented plants, *Lachnospiraceae* was identified as one of the top 20 microbes most strongly associated with the fermented plant consumption, as determined by Songbird differential abundance analysis⁸.

Additionally, Cotillard *et al.* analyzed 10,085 adult samples from AGP and found *Lachnospiraceae* significantly enriched in individuals with a plant-based diet, compared to Exclusion diet (ED) (low-carbohydrate diets with nearly no consumption of starchy foods or sweet products), Flexitarian diet (FL), Standard Western diet (SW), Health-Conscious Western diet (HW), respectively based on DESeq2 analysis⁹.

Given the prior evidence linking *Lachnospiraceae* to dietary plant consumption in the broader project these samples are part of, we selected this genome to explore whether specific genomic regions correlate with plant intake. We have expanded our explanation for this selection in lines 506-514. We believe that focusing on *Lachnospiraceae* is well-justified based on its consistent association with dietary plant consumption across multiple studies.

I also tried to run micov on my own samples. The pipeline did not complete successfully due to the following issues.

Author response: We greatly appreciate the reviewer attempting to run micov on their own data, and we apologize for any technical issues on applying the pipeline. Please kindly see our responses below to what we think may be the underlying issues.

6. Only some `cov.gz` files were generated. Most were empty, with no log or error to indicate what went wrong. The `.sam` files are not empty.

Author response: We apologize for an inability to construct a minimally reproducible example of the behavior the reviewer encountered, without more detail we regrettably must guess as to what may have occurred. On the assumption the reviewer is commenting on “micov compress”, we politely direct the reviewer to the unit tests for the underlying compression logic here (https://github.com/biocore/micov/blob/6d8d3d43868166bd354e1d9fe68511eb0e4f6093/micov/tests/test_io.py#L508), the code emits a warning about empty .sam files here (<https://github.com/biocore/micov/blob/6d8d3d43868166bd354e1d9fe68511eb0e4f6093/micov/cli.py#L171-L172>), and the code which raises an exception on unexpected buffers here (https://github.com/biocore/micov/blob/6d8d3d43868166bd354e1d9fe68511eb0e4f6093/micov/_io.py#L565). The underlying .sam parser (https://github.com/biocore/micov/blob/6d8d3d43868166bd354e1d9fe68511eb0e4f6093/micov/_io.py#L356; https://github.com/biocore/micov/blob/6d8d3d43868166bd354e1d9fe68511eb0e4f6093/micov/tests/test_io.py#L559) is just a wrapper for the read_csv method of Polars. We are unsure about how the reviewer’s data deviates from what the code is tested against, but we would be eager to identify and resolve the problem if the reviewer is willing and able to provide additional detail.

7. The README references the command `per-sample-group`, which is not implemented. I expect this is supposed to be `per-sample`.

Author response: We apologize for the disconnection between the reviewer’s install of micov and the README. This was a tricky bug for us to isolate, because the command "per-sample" is not part of the release and had actually been previously deleted, giving the appearance that the wrong version may have been inadvertently installed.

Upon further investigation, we discovered that the issue was the result of an unexpected change in the implicit behavior of the command line interface library click (<https://github.com/pallets/click>) used in micov.

Specifically, click now modifies commands ending in "_group", specifically stripping the "_group" suffix. This is new as of their May 10 2025 release (<https://github.com/pallets/click/releases/tag/8.2.0>, <https://github.com/pallets/click/issues/2322>, <https://github.com/pallets/click/pull/2604>).

To resolve this, we have pinned click to < 8.2 in our pyproject.toml and published a new release 2025.7, which restores the expected commands. The updated version has also been synchronized with PyPI: <https://pypi.org/project/micov/#history>.

The reviewer should now be able to view and use the `per-sample-group` command by installing the newest version of micov. To install the most updated version, users may run (this instruction is taken from the GitHub repository for easy reference):

```
...  
conda create -n micov python=3.12  
conda activate micov  
pip install micov  
...
```

At the end of stdout output, users should see confirmation:

```
...  
"Successfully installed click-8.1.8 micov-2025.7"  
...
```

Users can verify installation with

```
...  
micov - - help  
...
```

This should return:

```
...  
Usage: micov [OPTIONS] COMMAND [ARGS]...
```

micov: microbiome coverage.

Options:

--help Show this message and exit.

Commands:

binning	Bin genome positions and quantify read and...
compress	Compress BAM/SAM/BED mapping data.
consolidate	Consolidate coverage files into a Qiita-like...

```
extract-sample-presence  Extract variables for each described feature...
nonqiita-to-parquet      Aggregate BED3 files to parquet.
per-sample-group       Generate sample group plots and coverage data.
position-plot            Construct a single sample coverage plot.
qiita-coverage           Compute aggregated coverage from one or more...
qiita-to-parquet         Aggregate Qiita coverage to parquet.
````
```

We believe this update will allow the reviewer to view and use the `per-sample-group` command as described in README:

<https://github.com/biocore/micov/tree/main>.

8. While the parquet files were created successfully, 'per-sample' with '--parquet-coverage', '--plot', '--sample-metadata', '--sample-metadata-column', '--features-to-keep' and '--output' produced no files, without any logging or errors.

**Author response:** We suspect that this issue might be related to 7) since the "per-sample" command does not exist in the release of micov prior to manuscript submission. Following the release of micov 2025.7, we have re-tested our example commands in the README and verified they are working as expected.

If the issue persists with the latest version, we would be eager to help identify and address the root issues. A helpful starting point would be to compare the reviewer's input and intermediate files to the example files provided in the `./example` directory of the repository.

We thank the reviewer again for helping us improve the usability and robustness of the tool.

#### Minor Comments

Figure 2b: Inset text is too small.

**Author response:** We appreciate the reviewer for pointing this out. We have increased the font size of the insert text in Figure 2b. Please kindly see below for the updated Figure 2.

Figure 2: micov detects phenotypic relevant strain variation, captures changes in genome abundance at the level of a single genomic copy in wastewater, and exhibits sensitive detection in low biomass specimens. (a) A scaled position plot of

*P. copri* in human gut microbiome samples collected from subjects in the US/UK/Mexico stratified by presence/absence of region PC351. Sample groups are ordered by increasing sample size. Grey dotted gridlines are added as a visual aesthetic to help understand the data relationship to the genome coordinates on the y-axis; (b) Coverage presence in this region is associated with greater overall genome coverage, with supporting Kolmogorov-Smirnov statistics. Notably, overall coverage was not significantly different between the US and UK for individuals containing the region (KS test,  $stat=0.17$ ,  $p=0.0711$ ), nor was it if they both lacked the region (KS test,  $stat=0.12$ ,  $p=0.0495$ ; n.s. if corrected); (c) Common high effect size variables, and per-sample characterization of region presence/absence, were tested with PERMANOVA against Weighted UniFrac; (d-e) PCoAs of the weighted UniFrac distances colored by the region (d) and colored by country (e); (f) A receiver operator curve for a nested cross validated Random Forest classifier predicting presence/absence of PC351; (g) Coverage for region L682 in the Lachnospiraceae genome exhibiting differential coverage related to the diversity of plant consumption; (h) Detection of enteropathogenic *E. coli* (EPEC) at increasing levels of genome copies spiked into untreated wastewater with a number of water controls (Methods). All spike-in levels show statistically significant elevated cumulative coverage levels compared to the background. A low background amount of *E. coli* is expected in wastewater. Only samples with non-zero EPEC coverage are shown; (i) The cumulative coverage of *M. gnavus* from different tissue types surgically collected from Crohn's Disease patients; (j) with supporting Kolmogorov-Smirnov statistics. Set of statistics shown are those which reported an uncorrected  $p$ -value below 0.05, with correction by the Bonferroni procedure; asterisks denote corrected  $p$ -values below 0.05. All Kolmogorov-Smirnov and PERMANOVA tests are in Supplementary Table 2.

Figure 2h: Inset text is too small.

**Author response:** We have increased the font size of the insert text in Figure 2h.

The cumulative lines indicate that some samples have 0% coverage of the spike-in. Is that expected? What proportion had <5% coverage?

**Author response:** We appreciate the reviewer's observation. We would like to clarify that Figure 2h only displays samples with non-zero coverages, which we have now clarified in the updated figure legend. The rationale for this decision is that micov's primary focus is on examining how coverage patterns vary across groups for samples that exhibit some evidence of the reference genome. Therefore, all group lines in Figure 2h begin from non-zero values - the lowest being 0.002% coverage.

However, the reviewer raised a very good point and in response, we re-analyzed data including samples with zero EPEC coverage (32 samples). Please note that

the total number of control/non-control samples in the analysis below differs slightly from Figure 2h, as this expanded analysis includes samples with zero EPEC coverages.

29 out of 412 (7.0%) control samples (samples with no EPEC spike-ins) and 3 out of 305 (1.0%) non-control samples (samples with EPEC spike-ins) have 0% coverage of the spike-in. The reviewer is correct that some samples have zero coverage of the spike-in. This is expected and may be due to variable sample extraction, library prep, and sequencing quality during standard processing.

Additionally, 26.9% (111 out of 412) control samples and 21.3% (65 out of 305) non-control samples had <5% EPEC coverage. It should be noted that for higher copies of spike-in, this proportion increases significantly. For example, samples in the highest spike-in group, 80.0% (24 out of 30) samples had <5% EPEC coverage. We thank the reviewer for asking this insightful question, which prompted us to clarify the starting points in Figure 2h and improve the interpretability of this figure.

Figure 2i: Legend location obscures green line. Also inset text is too small.

**Author response:** We have increased the font size of all insert legend text and repositioned the legend in Figure 2i to avoid overlap. Please note the shape of the grey region has slightly changed due to the inherent randomness in the Monte Carlo simulation but the interpretation of the figure remains the same.

## REFERENCES (For responses only)

1. Aroney, S. T. N. *et al.* CoverM: read alignment statistics for metagenomics. *Bioinformatics* **41**, (2025).
2. Eren, A. M. *et al.* Anvi'o: an advanced analysis and visualization platform for 'omics data. *PeerJ* **3**, e1319 (2015).
3. Olm, M. R. *et al.* inStrain profiles population microdiversity from metagenomic data and sensitively detects shared microbial strains. *Nat Biotechnol* **39**, 727–736 (2021).
4. Zhao, C., Dimitrov, B., Goldman, M., Nayfach, S. & Pollard, K. S. MIDAS2: Metagenomic Intra-species Diversity Analysis System. *Bioinformatics* **39**, (2023).
5. Blanco-Miguez, A. *et al.* Extending and improving metagenomic taxonomic profiling with uncharacterized species with MetaPhlAn 4. *bioRxiv* (2022) doi:10.1101/2022.08.22.504593.
6. Truong, D. T., Tett, A., Pasolli, E., Huttenhower, C. & Segata, N. Microbial strain-level population structure and genetic diversity from metagenomes. *Genome Res* **27**, 626–638 (2017).
7. McDonald, D. *et al.* American Gut: an Open Platform for Citizen Science Microbiome Research. *mSystems* **3**, (2018).
8. Taylor, B. C. *et al.* Consumption of fermented foods is associated with systematic differences in the gut microbiome and metabolome. *mSystems* **5**, (2020).
9. Cotillard, A. *et al.* A posteriori dietary patterns better explain variations of the gut microbiome than individual markers in the American Gut Project. *Am J Clin Nutr* **115**, 432–443 (2022).

## Reprocessing Note

During the preparation of this response, we note an important change in upstream processing of the THDMI dataset, which, although it does not materially affect the results in this paper, we discovered was necessary while verifying the Qiita preparation IDs. Specifically, we learned the set of IDs reported in the THDMI paper (10.1128/msystems.00544-25) were incorrect. The preparation IDs described in that paper were host filtered using an out-of-date protocol rather than the protocol described in the methods text, and did not reflect the actual data used in that THDMI paper. Because of the error in the THDMI paper, we had inadvertently used the wrong data in this manuscript, which we have now corrected.

Specifically, we have taken the following two steps:

(1) For this paper, we reprocessed using the correct THDMI preparations. These preparations use a consistent and comprehensive host filtering procedure (10.1038/s41467-025-56077-5). We only observe minor differences in results, such as a rank order change for the top regions of interest for *P. copri*. We do not observe any changes that affect conclusions. The minor variation in results corresponds with our expectations, because human stool has a low proportion of human DNA, and for the small amount of human DNA that is present, we only anticipate recruitment to microbial genomes in cases of reference genome contamination with human sequence. Notably, and in support of the minor differences resulting from a change of host filtering, we reexamined the database masking outputs as performed in Sepich-Poore et al (10.1038/s41388-024-02974-w), and we do not observe regions of *P. copri* (G000157935) or *Lachnospiraceae* (G900066105) that contain putative human contamination.

(2) We have started the process of getting co-author approval for an erratum for the THDMI paper (10.1128/msystems.00544-25).

Please see below for the mapping information between the Qiita prep IDs incorrectly reported as used in the THDMI paper, and the correct Qiita prep IDs now used in this manuscript. The 'THDMI Data' section under Methods as well as Data Availability section have been updated accordingly.

| Incorrectly Reported in THDMI paper | Correct data used in micov manuscript |
|-------------------------------------|---------------------------------------|
| 10903                               | 16848                                 |
| 10915                               | 16854                                 |
| 10967                               | 16853                                 |
| 11594                               | 16849                                 |
| 11595                               | 16875                                 |

|       |       |
|-------|-------|
| 13279 | 16756 |
| 13280 | 16761 |
| 13358 | 16766 |

**Reviewer #1 (Remarks to the Author):**

The authors have addressed the suggestions and incorporated the requested modifications. I believe the manuscript can now be accepted for publication in its current form.  
Sincerely

We thank the reviewer for the positive feedback and for recognizing the revisions we have made. We are glad that the changes have addressed the reviewer's suggestions and that the reviewer considers the manuscript suitable for publication in its current form. We greatly appreciate the reviewer's time and effort in reviewing our work.

**Reviewer #2 (Remarks to the Author):**

Authors have adequately addressed all of my previous comments, and I now believe the manuscript is suitable for publication in Communications Biology.

We thank the reviewer for the positive evaluation. We are glad that our revisions have addressed all previous comments. We greatly appreciate the reviewer's time and constructive feedback throughout the review process.

**Reviewer #3 (Remarks to the Author):**

I appreciate the improved explanation of cumulative coverage. There is a lot of supporting analysis present only in the response to reviewers document. Are these going to be added to a supplemental document? I found the analysis in the response to Reviewer 2 point 10 particularly useful.

We thank the reviewer for this important point. We have included the response to Reviewer 2 point 10 in Supplementary Note 1 and cited it in the manuscript in Lines 217-220:

*This approach bypasses the selection of a coverage breadth threshold in detecting presence/absence of a particular microbial taxon in metagenomic samples (Supplementary Note 1).*

We greatly appreciate reviewer's feedback for improving the manuscript.

Most of my concerns have been adequately addressed, however there are a few that remain.

1. Do you also have benchmarking comparison for thread counts >1? Researchers are much more likely to use HPC systems for this type of analysis, so it would be useful to compare across a range of thread counts.

We thank the reviewer for this insightful suggestion. We agree that researchers are much more likely to use HPC systems for coverage breadth analysis, especially across hundreds of samples. Per the reviewer's suggestion, we ran micov on the largest test data ( $10^7$  alignment) with thread number ranging from one to 64 (**Table R1**).

**Table R1.** Runtime of micov on  $10^7$  alignments

| Thread number | Time Cost (s) |
|---------------|---------------|
| 1             | 61            |
| 2             | 54            |
| 4             | 71            |
| 8             | 93            |
| 16            | 99            |
| 32            | 114           |
| 64            | 137           |

Micov doesn't scale well with more threads on these data, with minor improvement at two threads and degradation at higher thread count. The reason for this is the test data used represents a single SAM file with varying number of alignments ( $10^2$ ,  $10^3$ ,  $10^4$ , ...,  $10^7$ ); micov is optimized for data rather than task level parallelism, and we have not yet optimized task level work. This design decision was taken because it exploits sample independence, allowing processing of many samples concurrently. The same mode of operation is not uncommon with CoverM, thus the current performance comparison does reflect common HPC use of micov and CoverM, where it encourages processing collections of samples using job arrays native to HPC job managers. Doing so, with micov, we regularly efficiently utilize hundreds or more cores within a HPC environment.

We thank the reviewer for raising this important question.

2. It is unclear why Figure R2 is not used in place of Figure 2H. From the response text, the difference appears to be the removal of water controls for the spike ins. Why these were included in the analysis is not explained. With their addition, Figure 2H does not represent realistic sample groups, and is therefore of little use for practical interpretation.

We thank the reviewer for understanding the distinction between Figure 2H and Figure R2, and we agree that it would be more appropriate to remove water controls from the relevant figure in the main manuscript. We have replaced Figure 2H with R2 and adapted the figure legend accordingly.

|                  |              | statistic | p-value  | corrected |
|------------------|--------------|-----------|----------|-----------|
| Mexico (absent)  | UK (absent)  | 0.27      | 8.34E-07 | 1.25E-05  |
| Mexico (absent)  | US (absent)  | 0.18      | 8.02E-04 | 1.20E-02  |
| UK (absent)      | US (absent)  | 0.12      | 4.34E-02 | 6.51E-01  |
| Mexico (present) | UK (present) | 0.31      | 2.62E-07 | 3.94E-06  |
| Mexico (present) | US (present) | 0.35      | 2.23E-09 | 3.35E-08  |
| UK (present)     | US (present) | 0.17      | 7.11E-02 | 1.00E+00  |

**c**

|                                | statistic | p-value  | corrected |
|--------------------------------|-----------|----------|-----------|
| PC351                          | 78.66     | 1.00E-06 | 2.80E-05  |
| Country                        | 63.20     | 1.00E-06 | 2.80E-05  |
| Country_PC351                  | 38.15     | 1.00E-06 | 2.80E-05  |
| Bowel Movement Quality_PC351   | 10.60     | 1.00E-06 | 2.80E-05  |
| Antibiotic History_PC351       | 10.38     | 1.00E-06 | 2.80E-05  |
| Types of Plants_PC351          | 8.41      | 1.00E-06 | 2.80E-05  |
| Bowel Movement Frequency_PC351 | 7.35      | 1.00E-06 | 2.80E-05  |
| Bowel Movement Quality         | 2.92      | 8.44E-04 | 2.36E-02  |
| Antibiotic History             | 2.47      | 3.78E-03 | 1.06E-01  |
| Bowel Movement Frequency       | 2.21      | 2.88E-03 | 8.06E-02  |
| Types of Plants                | 1.45      | 9.80E-02 | 1.00E+00  |

**j**

|                   |                              | statistic | p-value  | corrected |
|-------------------|------------------------------|-----------|----------|-----------|
| Involved Mucosa   | Mesenteric Fat               | 0.32      | 4.19E-03 | 1.17E-01  |
| Involved Mucosa   | Subcutaneous Fat             | 0.52      | 5.82E-06 | 1.63E-04  |
| Involved Mucosa   | Uninvolved Mucosa            | 0.33      | 6.49E-03 | 1.82E-01  |
| Involved Mucosa   | Creeping Fat                 | 0.40      | 8.92E-04 | 2.50E-02  |
| Involved Mucosa   | Distant Mesenteric Fat       | 0.37      | 3.23E-02 | 9.04E-01  |
| Involved Mucosa   | Negative control             | 0.53      | 4.20E-06 | 1.18E-04  |
| Involved Mucosa   | Monte Carlo unfocused (n=68) | 0.31      | 6.70E-03 | 1.88E-01  |
| Mesenteric Fat    | Subcutaneous Fat             | 0.37      | 1.19E-03 | 3.32E-02  |
| Mesenteric Fat    | Negative control             | 0.33      | 7.18E-03 | 2.01E-01  |
| Subcutaneous Fat  | Uninvolved Mucosa            | 0.29      | 3.31E-02 | 9.27E-01  |
| Uninvolved Mucosa | Negative control             | 0.33      | 1.09E-02 | 3.06E-01  |

**Modified Figure 2:** (caption see below)

Additionally, you state that it only includes samples with non-zero coverage. It would be helpful to have the number of samples with non-zero coverage stated in the legend, to give an idea of how many samples were necessary to get detection of the claimed “one genomic copy”.

We thank the review for this helpful suggestion, and we have added the number of samples (562) with non-zero coverage into the figure legend. The updated figure legend is:

**Modified Figure 2 (Lines 706-731):** *micov detects phenotypic relevant strain variation, captures changes in genome abundance at the level of a single genomic copy in wastewater, and exhibits sensitive detection in low biomass specimens. (a) A scaled position plot of *P. copri* in human gut microbiome samples collected from subjects in the US/UK/Mexico stratified by presence/absence of region PC351. Sample groups are ordered by increasing sample size. Grey dotted gridlines are added as a visual aesthetic to help understand the data relationship to the genome coordinates on the y-axis; (b) Coverage presence in this region is associated with greater overall genome coverage, with supporting Kolmogorov-Smirnov statistics. Notably, overall coverage was not significantly different between the US and UK for individuals containing the region (KS test,  $stat=0.17$ ,  $p=0.0711$ ), nor was it if they both lacked the region (KS test,  $stat=0.12$ ,  $p=0.0434$ ; n.s. if corrected); (c) Common high effect size variables, and per-sample characterization of region presence/absence, were tested with PERMANOVA against Weighted UniFrac; (d-e) PCoAs of the weighted UniFrac distances colored by the region (d) and colored by country (e); (f) A receiver operator curve for a nested cross validated Random Forest classifier predicting presence/absence of PC351; (g) Coverage for region L682 in the *Lachnospiraceae* genome exhibiting differential coverage related to the diversity of plant consumption; (h) Detection of enteropathogenic *E. coli* (EPEC) at increasing levels of genome copies spiked into untreated wastewater (Methods). All spike-in levels show statistically significant elevated cumulative coverage levels compared to the background. A low background amount of *E. coli* is expected in wastewater. Only samples with non-zero EPEC coverage ( $n = 562$ ) are shown; (i) The cumulative coverage of *M. gnavus* from different tissue types surgically collected from Crohn’s Disease patients; (j) with supporting Kolmogorov-Smirnov statistics. Set of statistics shown are those which reported an uncorrected p-value below 0.05, with correction by the Bonferroni procedure; asterisks denote corrected p-values below 0.05. All Kolmogorov-Smirnov and PERMANOVA tests are in Supplementary Table 2.*

Additionally, we have clarified in the Methods section that water controls were excluded in Figure 2h (Line 559):

*Spike-in’s were performed at the level of extracted wastewater gDNA and some control samples spiked-in to only water (154 samples), directly prior to sequencing library preparation. Water controls were excluded from Figure 2h.*

We appreciate the reviewer’s help in improving the clarity of the manuscript.

Previously, I installed micov using its conda version (2025.2), but since that hasn’t been

updated, I switched to installing via pip. I have now run the new version of micov using my own samples. Thank you for fixing the bugs. I especially appreciate that fixing the bug caused by the click library must have been frustrating.

We thank the reviewer for testing the updated version of micov. We confirm that the PyPI distribution has been kept up to date. We apologize for the delay in the update to conda-forge. The automatic procedure that sources from PyPI was blocked due to a minor difference in version pins in the conda recipe. We have now updated the conda-forge recipe and observe 2025.9 as available:

```
$ conda search -c conda-forge micov
Loading channels: done
Name Version Build Channel
micov 2025.2 pyhd8ed1ab_0 conda-forge
micov 2025.9 pyhd8ed1ab_0 conda-forge
```

Regarding the click library bug, we appreciate the reviewer's understanding. We are glad to hear the new version worked successfully with the reviewer's samples.